# 4-Hydroxy-2-nonenal antimicrobial toxicity is neutralized by an intracellular pathogen

Hannah Tabakh[1], Adelle P McFarland[1,2], Maureen K Thomason[1], Alex J Pollock[1], Rochelle C Glover[1], Shivam A Zaver[1,2], Joshua J Woodward[1]*

[1]Department of Microbiology, University of Washington, Seattle, United States; [2]Molecular and Cellular Biology Program, University of Washington, Seattle, United States

**Abstract** Pathogens encounter numerous antimicrobial responses during infection, including the reactive oxygen species (ROS) burst. ROS-mediated oxidation of host membrane poly-unsaturated fatty acids (PUFAs) generates the toxic alpha-beta carbonyl 4-hydroxy-2-nonenal (4-HNE). Although studied extensively in the context of sterile inflammation, research into 4-HNE's role during infection remains limited. Here, we found that 4-HNE is generated during bacterial infection, that it impacts growth and survival in a range of bacteria, and that the intracellular pathogen *Listeria monocytogenes* induces many genes in response to 4-HNE exposure. A component of the *L. monocytogenes* 4-HNE response is the expression of the genes *lmo0103* and *lmo0613,* deemed *rha1* and *rha2* (**r**eductase of **h**ost **a**lkenals), respectively, which code for two NADPH-dependent oxidoreductases that convert 4-HNE to the product 4-hydroxynonanal (4-HNA). Loss of these genes had no impact on *L. monocytogenes* bacterial burdens during murine or tissue culture infection. However, heterologous expression of *rha1/2* in *Bacillus subtilis* significantly increased bacterial resistance to 4-HNE in vitro and promoted bacterial survival following phagocytosis by murine macrophages in an ROS-dependent manner. Thus, Rha1 and Rha2 are not necessary for 4-HNE resistance in *L. monocytogenes* but are sufficient to confer resistance to an otherwise sensitive organism in vitro and in host cells. Our work demonstrates that 4-HNE is a previously unappreciated component of ROS-mediated toxicity encountered by bacteria within eukaryotic hosts.

*For correspondence:
jjwoodwa@uw.edu

Competing interests: The authors declare that no competing interests exist.

## Introduction

Innate immune detection of bacterial infection initiates a complex inflammatory response characterized by production of cytokines and small molecule mediators involved in driving antimicrobial immunity. A key aspect of intrinsic cellular immunity is the production of highly reactive molecules, including reactive oxygen (ROS) and nitrogen (RNS) species (*Nathan and Cunningham-Bussel, 2013*). ROS and RNS encompass a broad group of distinct molecules, including nitric oxide, hydrogen peroxide, hypochlorite, and superoxide, among others. Unlike adaptive immune responses, which are highly specific toward infectious agents, ROS and RNS exhibit broad toxicity toward biological systems through their capacity to react with lipid, amino acid, and nucleic acid moieties that are conserved among both eukaryotic hosts and invading microbes (*Patel et al., 1999*; *Jacobson, 1996*). While such indiscriminate noxious metabolite production provides protection against infection, many bacterial pathogens have evolved a diverse array of mechanisms to directly detoxify or repair damaged cellular components following ROS and RNS encounters, such as superoxide dismutases, catalases, peroxidases, and nitric oxide reductases (*Fang, 2004*; *Staerck et al., 2017*).

While many of the distinct chemical agents that comprise ROS and RNS are well characterized molecular components of the innate immune response, these molecules give rise to numerous secondary metabolites that may also contribute to host defense against infection. An initial characteristic of the inflammatory response is the mobilization of arachidonic acid from cellular membranes. Although arachidonic acid is most commonly thought of as the chemical precursor of eicosanoids, upon exposure to oxygen radicals derived from the ROS burst it undergoes a peroxide-mediated structural rearrangement, leading to the generation of breakdown products, the best studied and most abundant of which is 4-hydroxy-2-nonenal (4-HNE), a highly reactive membrane-permeable molecule (*Hanna and Hafez, 2018*). Over the last 40 years, the production of 4-HNE has been well documented at sites of sterile inflammation and has been associated with many disease pathologies, including atherosclerosis (*Uchida et al., 1994*), Alzheimers' (*Sayre et al., 2002*), diabetes (*Pillon et al., 2012*), obstructive pulmonary disease (*Rahman et al., 2002*), and chronic liver disease (*Paradis et al., 1997*).

4-HNE's toxicity is driven by its highly reactive αβ-unsaturated aldehyde, which is subject to both Michael addition and electrophilic addition to the aldehyde. 4-HNE is thus highly reactive against all nucleophilic moieties present in the cell, including amino acids, nucleotides and lipids (*Dalleau et al., 2013*). To combat this reactivity, eukaryotic organisms utilize detoxification enzymes including oxidoreductases that reduce the carbon-carbon double bond to generate 4-hydroxynonanal (4-HNA) (*Wang et al., 2019*; *Srivastata et al., 1996*; *Dick et al., 2001*; *Mano et al., 2002*), aldo-keto reductases that reduce the carbonyl group, forming the alcohol 1,4-dihydroxynonene (1,4-DHN) (*Hartley et al., 1995*), and aldehyde dehydrogenases and P450s that oxidize the carbonyl bond to the corresponding carboxylic acid and 4-hydroxynonenic acid (4-HNEA) (*Amunom et al., 2007*; *Guéraud, 2017*). Michael addition by glutathione, a reaction that occurs spontaneously and is catalyzed by glutathione-S-transferases, forms glutathionyl-4-hydroxynonenal (GS-HNE), which is reduced to glutathionyl-1,4-dihydroxynonene (GS-DHN) by aldose reductases (*Ramana et al., 2006*). In addition to enzymatic detoxification, 4-HNE toxicity can be ameliorated non-enzymatically through buffering agents, including quenching reactions with the endogenous peptides carnosine and GHK (Gly-His-Lys), as well as the small molecule hydrogen sulfide ($H_2S$) (*Mol et al., 2017*). Although reactive oxygen species generation and arachidonic acid mobilization and detoxification are well-known and well-studied components of innate immune responses, detailed studies characterizing the role of 4-HNE during infectious disease, particularly in the context of bacterial pathogens, are lacking.

In this study, we demonstrate that 4-HNE is generated during bacterial infection both in cell culture and in vivo. This mammalian metabolite is able to penetrate the bacterial cell envelope and access the cytoplasm, leading to bacterial growth delay or death. We observed that relative to a variety of bacterial species, the intracellular bacterial pathogen *Listeria monocytogenes* is highly resistant to the bactericidal effects of 4-HNE and that a broad transcriptional response is induced by toxic 4-HNE exposure, including two genes *lmo0103* and *lmo0613*, deemed *rha1* and *rha2* (**r**eductase of **h**ost **a**lkenals), respectively. The loss of both *rha1* and *rha2* sensitizes *L. monocytogenes* to 4-HNE toxicity. Through in vitro analysis of recombinant Rha1 and Rha2, we found both enzymes reduce 4-HNE in an NADPH-dependent-manner to the saturated aldehyde 4-HNA. Importantly, when *rha1/2* are expressed in the 4-HNE sensitive and avirulent organism *B. subtilis*, they significantly increase bacterial survival in the presence of 4-HNE in vitro and following phagocytosis by murine macrophages in a manner dependent upon ROS generation. Our findings are consistent with the premise that 4-HNE is a heretofore unrecognized component of ROS-toxicity encountered by bacteria during infection and that detoxification mechanisms used to counteract 4-HNE-mediated cytotoxicity facilitate bacterial survival within eukaryotic hosts.

## Results

### 4-HNE accumulates during *L. monocytogenes* infection

4-HNE is a highly reactive electrophilic αβ-unsaturated aldehyde that undergoes Michael addition with nucleophilic amino acids, resulting in stable conjugates that correlate with cellular levels of free 4-HNE. Monoclonal antibodies to these adducts are routinely used to monitor 4-HNE levels in cells (*Majima et al., 2002*). To investigate 4-HNE production during bacterial infection, we infected

murine hepatocytes with *L. monocytogenes* for 6 hr and quantified 4-HNE protein conjugates using dot blots of whole cell lysates at various times post-infection. As a control, we also quantified adducts that accumulated after treating uninfected cells with 10 μM pure 4-HNE. We found that at 6 hr post infection, 4-HNE adducts accumulate to a similar level as those observed with the addition of the pure compound (*Figure 1A*). To interrogate the impact of bacterial infection on host production of 4-HNE in vivo, mice were infected intravenously with *L. monocytogenes* constitutively expressing GFP. At 48 hr post infection, tissues were harvested, fixed, and analyzed by immunohistochemistry. Clear foci of infection were visible in the liver with no change in the abundance of 4-HNE protein conjugates (*Figure 1—figure supplement 1A–D*). In the spleen, however, bacteria were diffusely distributed throughout the organ and the entire spleen of infected mice exhibited increased staining for 4-HNE protein conjugates (*Figure 1B–E*).

At higher levels of magnification, we observed that 4-HNE conjugates within the spleen were not evenly distributed among all cells. Most of the tissue exhibited diffuse and constant staining and a subset of cells showed very dark and robust staining for 4-HNE conjugates (*Figure 1E*). At ×100 magnification, the most pronounced signal for *L. monocytogenes* exhibited punctate staining (*Figure 1F*) and a similar pattern was observed for 4-HNE conjugates at this magnification (*Figure 1G*). While these observations do not provide quantitative measures of 4-HNE levels, they establish that 4-HNE was indeed elevated in the spleen following bacterial infection and suggest that bacteria encounter this metabolite within the host.

## 4-HNE causes damage through the targeting of nucleophilic protein moieties and *L. monocytogenes* is resistant to 4-HNE-mediated death

Electrophilic stress due to 4-HNE conjugation to proteins causes eukaryotic cells to undergo apoptosis following intermediate 4-HNE exposure (5–40 μM) and necrosis at higher concentrations (40–100 μM) (*Dalleau et al., 2013*). However, due to 4-HNE's lipophilicity it is believed to accumulate to significantly higher levels (0.3–5 mM) near and within membranes than what is typically considered cytotoxic (*Zimniak, 2011*; *Esterbauer et al., 1991*; *Uchida, 2003*).

To characterize bacterial sensitivity to 4-HNE toxicity, we exposed a panel of both Gram-positive and Gram-negative bacteria to a wide range of 4-HNE concentrations and assessed viability. We observed variability in survival, ranging from a 4-log reduction in CFU for *B. subtilis* and *Francisella novicida*, a 2-log reduction for *Staphylococcus aureus*, a 1-log reduction for *Escherichia coli* and *Pseudomonas aeruginosa*, to a half-log reduction for *Enterococcus faecalis*. For *L. monocytogenes*, less than a half-log reduction was observed up to 640 μM of 4-HNE (*Figure 2A*). The variability in survival did not appear to track with bacterial phylum or their cellular infection cycle, as *L. monocytogenes* and *F. novicida*, both intracellular pathogens, had markedly different survival capabilities following 4-HNE exposure.

The significant resistance of *L. monocytogenes* to 4-HNE exposure was striking. Although 4-HNE exhibited limited bactericidal activity toward this organism, we observed a dose-dependent delay in growth of *L. monocytogenes* with increased exposure to 4-HNE (*Figure 2B*). Due to the conserved nature of 4-HNE targets, we hypothesized that 4-HNE would exert similar damaging effects on bacteria as on eukaryotic cells. Thus, we first interrogated the ability of 4-HNE to generate protein adducts within *L. monocytogenes*. Dot blots of *L. monocytogenes* cell lysates from bacteria exposed to increasing concentrations of 4-HNE indicated an accumulation of 4-HNE-protein adducts that correlated with increased 4-HNE exposure (*Figure 2C*), establishing that this aldehyde penetrates the bacterial cell envelope and impacts cytosolic proteins.

4-HNE adduct accumulation can result in protein misfolding and crosslink-induced aggregation. Eukaryotic cells clear 4-HNE damaged proteins through proteasome and autophagy-mediated pathways (*Zhang and Forman, 2017*). Bacteria target damaged proteins for degradation through the proteases that comprise the heat shock response (*Parsell and Lindquist, 1993*). Because this response is primarily transcriptionally regulated (*Yura et al., 1993*), RT-qPCR was performed on a subset of genes representing two major groups of heat shock genes in *L. monocytogenes*: HrcA-regulated chaperones and CtsR-regulated proteases (*Roncarati and Scarlato, 2017*). When *L. monocytogenes* was exposed to 640 μM 4-HNE for 20 min, the four genes tested (*clpC, clpE, dnaK, groES*) were induced 3-to-40-fold compared to vehicle controls (*Figure 2D*). These data, combined with the dot blot results, support the hypothesis that 4-HNE causes protein damage to which *L. monocytogenes* mounts a heat shock response. These observations are consistent with previous reports of

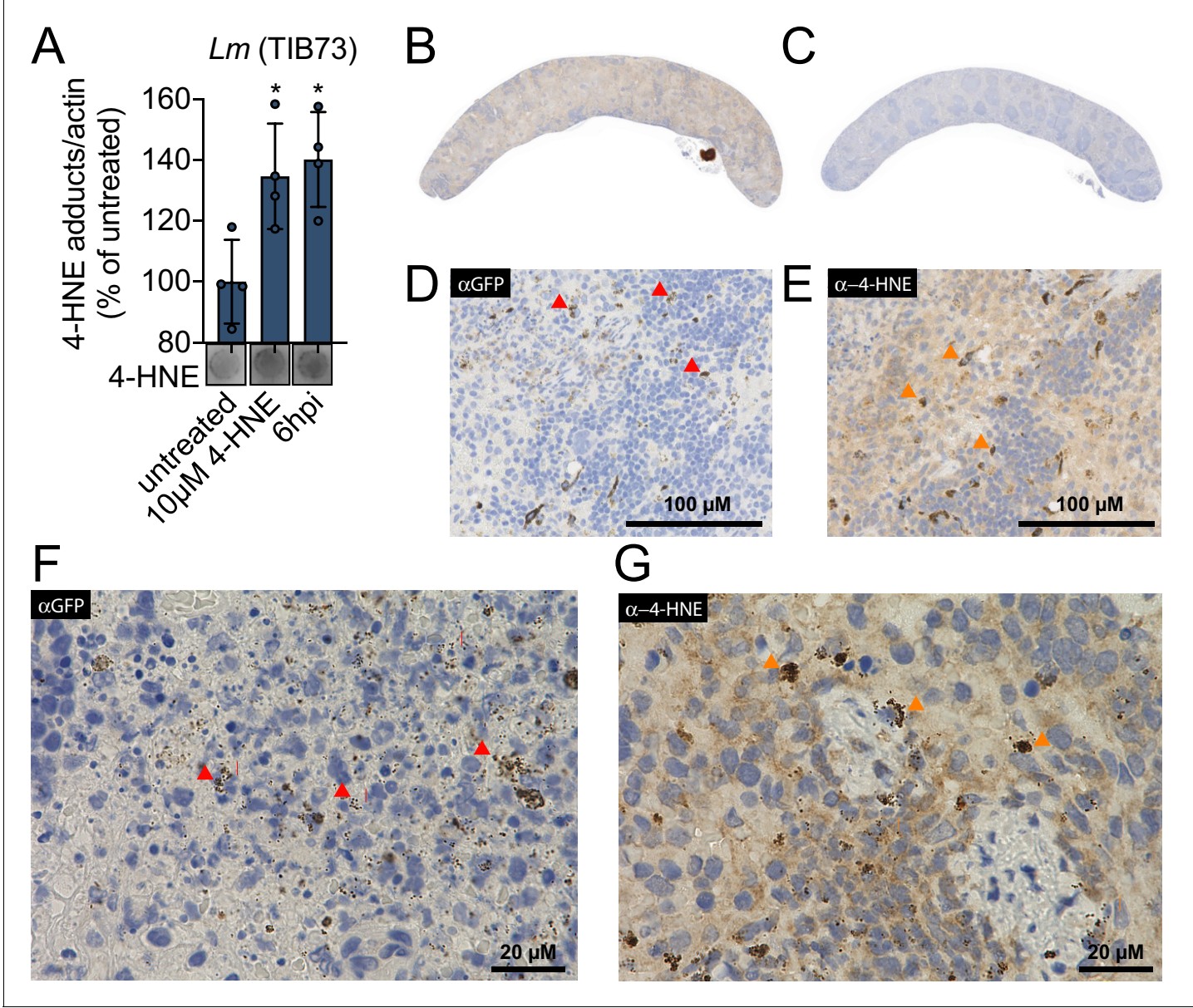

**Figure 1.** 4-HNE accumulates during intracellular bacterial infection by *L. monocytogenes*. (**A**) 4-HNE accumulation in TIB73 murine hepatocytes during intracellular *L. monocytogenes* infection. 4-HNE adduct accumulation was assessed by dot blot of whole cell lysates normalized to actin levels. Data are normalized 4-HNE/actin levels as percent of 4-HNE/actin in untreated sample. Dot blot image below is representative. (**B**) 4-HNE accumulation in the spleen after 48 hr murine infection by GFP⁺ *L. monocytogenes* assessed by immunohistochemistry analysis with anti-4-HNE antibody. (**C**) Uninfected spleen with anti-4-HNE antibody. (**D**) Infected spleen at ×25 magnification with anti-GFP antibody. (**E**) Infected spleen at ×25 magnification with anti-4-HNE antibody. (**F**) Infected spleen at ×100 magnification with anti-GFP antibody. (**G**) Infected spleen at ×100 magnification with anti-4-HNE antibody. Red arrows in D and F indicate *L. monocytogenes* (GFP) detection in the tissue. Orange arrows in E and G indicate cells with concentrated 4-HNE staining. Antigens were detected with 3,3-diaminobenzidine staining by horseradish peroxidase and cellular nuclei imaged with Hematoxylin counterstain in panels B-G. Data in (**A**) are in biological quadruplicate. Statistics in (**A**) are an ordinary one-way ANOVA with a Dunnett's multiple comparison test against untreated. Error bars are mean ± SD. *p<0.05.

The online version of this article includes the following figure supplement(s) for figure 1:

**Figure supplement 1.** 4-HNE does not accumulate in the liver during infection by *L. monocytogenes*.

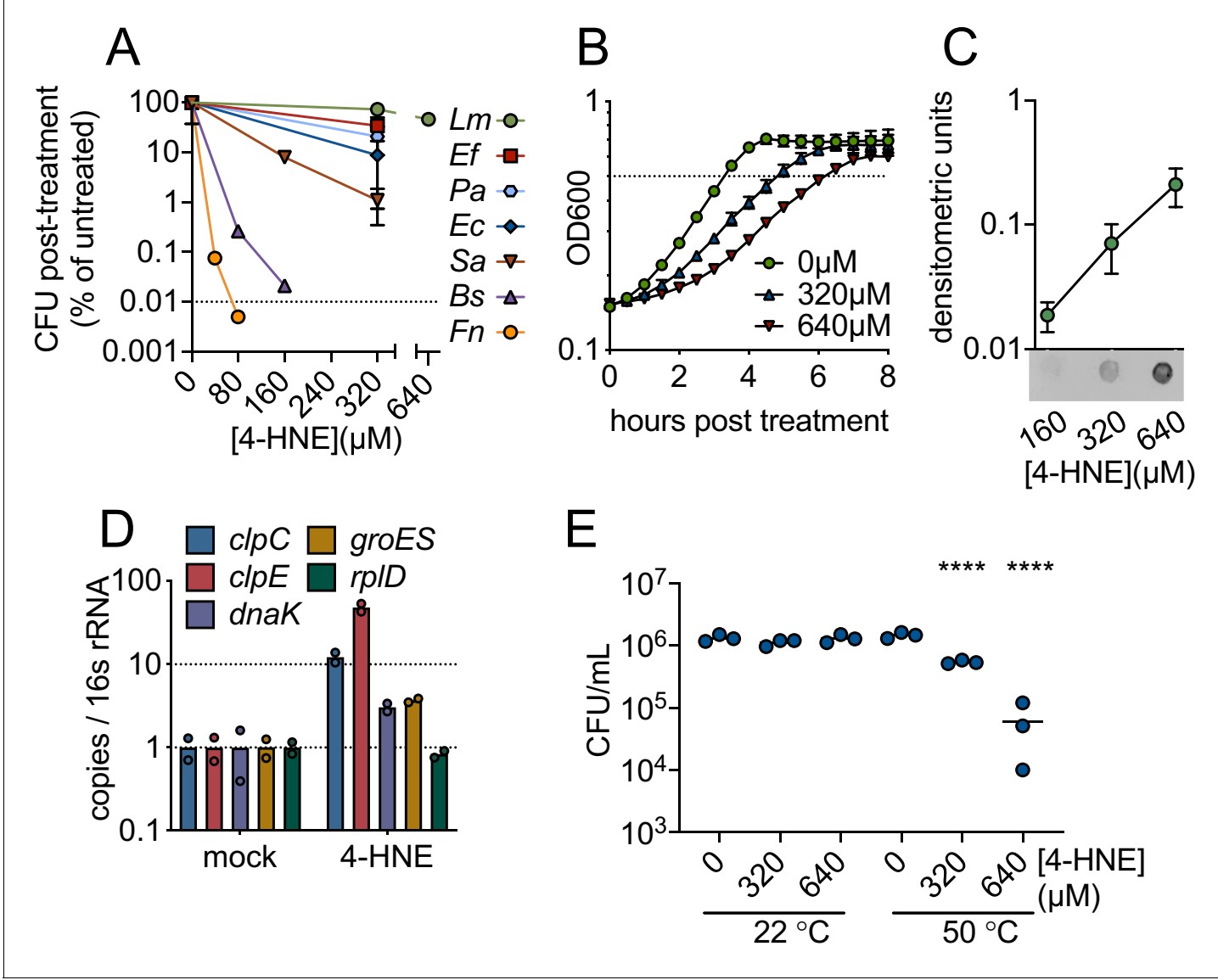

**Figure 2.** 4-HNE is a bactericidal, cell-permeable and protein damaging molecule. (A) Survival of mid-log (0.4–0.8 $OD_{600}$) *Listeria monocytogenes (Lm)*, *Enterococcus faecalis (Ef)*, *Pseudomonas aeruginosa (Pa)*, *Escherichia coli (Ec)*, *Staphylococcus aureus (Sa)*, *Bacillus subtilis (Bs)*, and *Francisella novicida (Fn)* following exposure to various concentrations of 4-HNE or mock vehicle (ethanol) in PBS at 37°C for 1 hr. Data are reported as recovered CFU normalized to mock-treated controls. Dashed line is at the limit of detection. (B) Growth of *L. monocytogenes* in TSB at 37°C with various concentrations of 4-HNE added at time zero. Dashed line at $OD_{600}$ 0.5 (C) Anti-4-HNE dot blot of soluble bacterial lysates from mid-log *L. monocytogenes* resuspended in PBS and treated with increasing concentrations of 4-HNE for 30 min. Protein levels were normalized (3 μg total protein) and signal quantified by densitometry on a Licor Odyssey Fc. (D) RT-qPCR measurement of expression of the indicated genes from mid-log *L. monocytogenes* in TSB treated with 640 μM 4-HNE for 20 min. Expression normalized to 16S rRNA levels. (E) Recovered CFU of WT *L. monocytogenes* following exposure to 4-HNE or mock vehicle (ethanol) in PBS at 37°C for 1 hr, followed by heat shock (50°C) or no heat shock (22°C) treatment for 10 min. Data in figures (A) and (B) are in biological triplicate. Data in (C) and (D) are biological duplicate. Data in (E) are in technical triplicate and representative of at least two independent experiments. Statistics in (E) are an ordinary one-way ANOVA with a Dunnett's multiple comparison test against each untreated. Error bars are mean ± SD. *p<0.05; **p<0.01; ***p<0.001; ****p<0.0001. In figure (E), the line is drawn at the median of data.

electrophile stress induced expression of Clp proteases in *B. subtilis* (*Nguyen et al., 2009*). To further explore this connection, bacteria were treated with increasing concentrations of 4-HNE prior to a sublethal heat shock (50°C for 10 min) and bacterial survival was determined by CFU analysis. Consistent with 4-HNE-induced proteotoxic stress, elevated levels of 4-HNE exposure sensitized the bacteria to heat (*Figure 2E*). The elevated induction of cellular proteases relative to chaperones may indicate that *L. monocytogenes* primarily combats electrophilic 4-HNE stress through turnover of

damaged proteins rather than chaperone-mediated stabilization. Collectively, these observations suggest that despite the formation of protein adducts and delayed growth following exposure, *L. monocytogenes* has a robust capacity to survive 4-HNE toxicity, although its exposure sensitizes this organism to other proteotoxic stressors.

## *L. monocytogenes* expresses potential 4-HNE detoxification enzymes

Our data suggest that *L. monocytogenes* is exposed to 4-HNE during infection and that only high concentrations of this aldehyde impact its growth. We hypothesized that *L. monocytogenes* may express genes involved in countering the cytotoxic effects of 4-HNE. To probe further, we performed global transcriptome analysis during 4-HNE exposure using RNA sequencing. Over one hundred genes were induced greater than 10-fold in response to 4-HNE exposure, including several of the heat shock genes previously identified by RT-qPCR analysis (*Figure 3A*, *Figure 3—figure supplement 1*, *Source data 1*).

Eukaryotic cells utilize several reductases to detoxify 4-HNE (*Mol et al., 2017*). Two reductases were highly induced in our global transcriptome, *lmo0103* and *lmo0613*, which we refer to as *rha1* and *rha2* (**r**eductase of **h**ost **a**lkenals 1 and 2), respectively. Rha1 is annotated as a nitroreductase. Phyre2 analysis of Rha1 predicted high structural homology to CLA-ER (PDB: 4QLY), a flavin-dependent enone reductase from *Lactococcus plantarum* (*Hou et al., 2015*; *Kelley et al., 2015*). Rha2 is annotated as an alcohol/quinone reductase. A Phyre2 analysis of Rha2 revealed structural similarity to crotonyl-CoA carboxylase/reductases (PDBs: 3KRT, 4Y0K, and 5A3J) and a plant chloroplast oxoene reductase (PDB: 5A3V), two enzymes with the capacity to reduce enone-containing lipophilic substrates. Given the predicted reductase activity of Rha1 and Rha2 and their structural similarity to proteins that metabolize αβ-unsaturated carbonyl-containing compounds, we further investigated their role in 4-HNE resistance.

Induction of *rha1* and *rha2* in response to 4-HNE exposure was found to be 34 and 90-fold, respectively, compared to untreated control, by RT-qPCR (*Figure 3B*). We subsequently exposed *L. monocytogenes* to a panel of aldehydes at equimolar concentrations that did not reduce CFU (*Figure 4—figure supplement 1A,B*; *Figure 3—figure supplement 2A,B*). This panel included 4-HNE; 4-HHE (4-hydroxy-2-hexenal), a similar but shorter chain αβ-unsaturated aldehyde produced from the oxidation of ω−3 fatty acids (*Awada et al., 2012*); methylglyoxal, a reactive byproduct of glycolysis; propionaldehyde, a short chain saturated aldehyde; and malondialdehyde, another product of lipid peroxidation (*Esterbauer et al., 1991*). Both *rha1* and *rha2* were most strongly induced by 4-HNE exposure, with much less induction by 4-HHE and negligible induction with the other tested compounds (*Figure 3B*), suggesting that their induction may be specific to this aldehyde.

To gain further insight into the regulation of these genes, we compared expression in a variety of stressors, including diamide-dependent disulfide stress, heat, and nitric oxide stress. Under all conditions tested the control gene *rplD* was unchanged (*Figure 3C*) and none of the conditions tested led to a reduction of bacterial CFU (*Figure 4—figure supplement 1C*; *Figure 3—figure supplement 2C*). We found that diamide induced *rha1* by 11-fold and *rha2* by 100-fold and both *clpC* and *groES* were induced by 70 and 40-fold, respectively. Heat shock induced both *rha1* and *rha2* by approximately 30-fold, comparable to the control genes *clpC* (20-fold) and *groES* (40-fold). NO was unable to induce either gene under the conditions tested, while the positive control gene *srtB* was induced approximately 25-fold (*Leichert et al., 2003*; *Richardson et al., 2006*). However, among the various stressors tested, the most robust induction of both *rha1* and *rha2* was with 4-HNE (300 and 500-fold, respectively), as we observed previously with the panel of aldehydes (*Figure 3B,C*).

We next assessed if *rha1* and *rha2* are induced by *L. monocytogenes* during intracellular infection. At 6 hr post infection of J774 macrophages, we found that there was significant induction of both genes compared to growth in BHI broth (*Figure 3D*). Together these transcriptional studies suggested that the *rha1/2* genes are robustly induced in response to 4-HNE and to a lesser extent by other aldehydes or cellular stresses. The induction of *rha2* by diamide suggests that *rha1* and *rha2* may be components of distinct stress regulons, with the latter also being involved in the disulfide stress response. However, 4-HNE is known to be reactive toward redox buffering thiols such as glutathione and therefore *rha2* may play a role in both responses.

These intriguing transcriptional results suggested that *rha1* and *rha2* may function in mediating 4-HNE resistance. To test this, we generated individual and double mutants of *rha1* and *rha2* and assessed survival of these mutants by competition experiments relative to WT *L. monocytogenes*

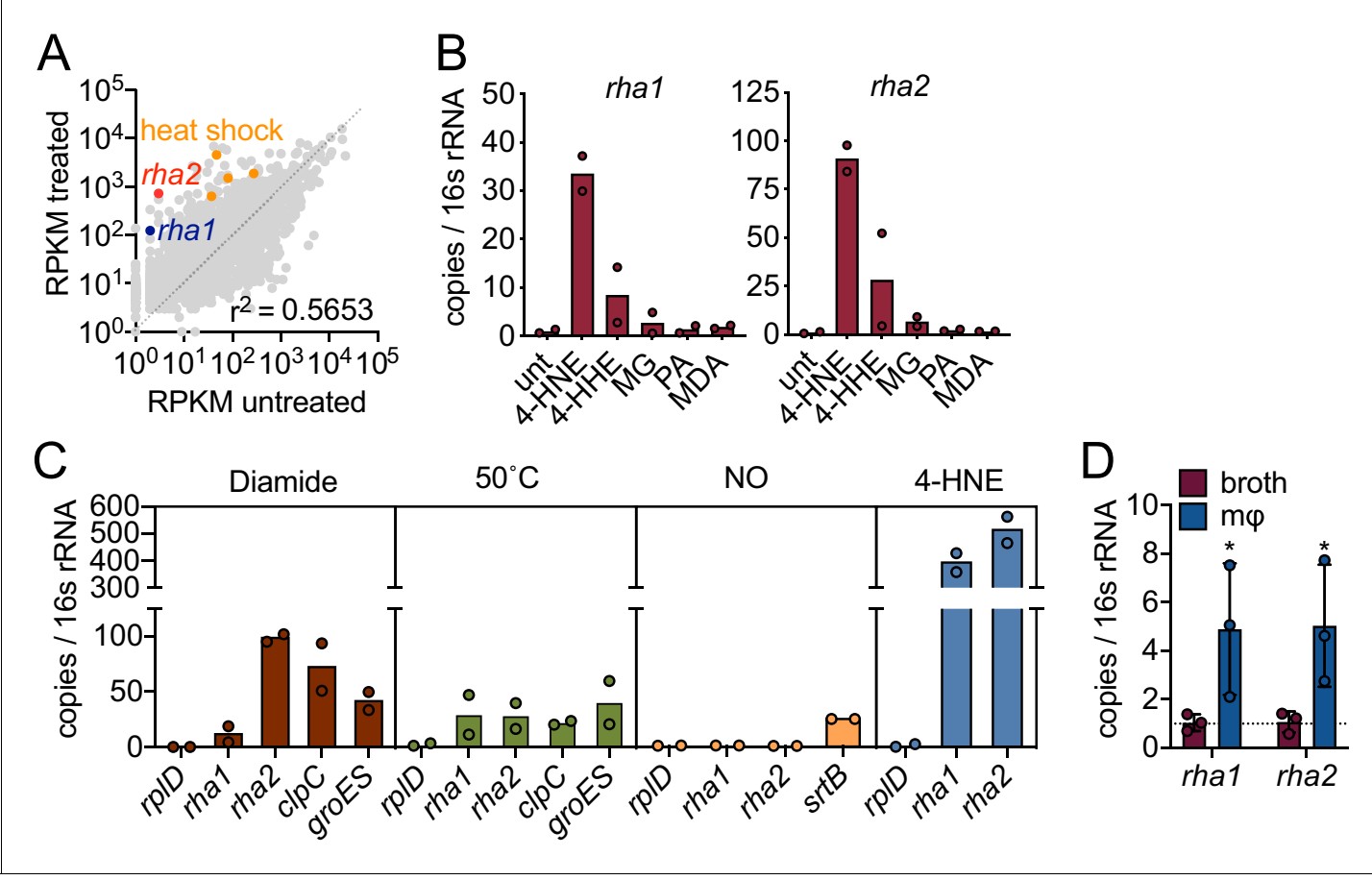

**Figure 3.** 4-HNE exposure induces resistance genes in *L. monocytogenes.* (A) Global gene expression of mid-log *L. monocytogenes* in TSB treated with 640 μM 4-HNE or ethanol control for 20 min. RPKM: reads per kilobase million. Genes of interest *rha1*, *rha2*, and heat shock class members are indicated in blue, red, and orange, respectively. (B) RT-qPCR of expression of *rha1* and *rha2* genes after 20 min treatment of mid-log bacteria in TSB media with 500 μM of selected aldehydes: 4-HNE (4-hydroxy-2-nonenal), 4-HHE (4-hydroxy-2-hexenal), MG (methylglyoxal), PA (propionaldehyde), and MDA (malondialdehyde). (C) RT-qPCR analysis of expression of *rha1* and *rha1* treated with sublethal levels of diamide (5 mM), heat (50°C), 4-HNE (640 μM), and nitric oxide (1 mM of the NO donor DEA/NO) for 20 min in TSB media. (D) RT-qPCR analysis of expression of *rha1* and *rha2* at 6 hr post infection in J774 macrophages (mφ). Data in figures (A) and (B) are in biological duplicate. (C) are two independent experiments, with two pooled biological duplicates within each experiment. (D) is in biological triplicate. The bar graphs in (B) (C) and (D) represent the mean of the data. Statistics in (D) are unpaired t-test between the ΔCt values of broth and macrophage samples. Error bars are mean ± SD. *p<0.05.

The online version of this article includes the following figure supplement(s) for figure 3:

**Figure supplement 1.** Global gene expression of mid-log *L. monocytogenes* in TSB treated with ethanol mock control (panel 1) or 640 μM 4-HNE (panel 1) for 20 min.

**Figure supplement 2.** Various aldehydes and chemical stresses and impacts on *L. monocytogenes* survival.

following 4-HNE exposure. Control mixtures left untreated in PBS exhibited no significant difference in competitive index between mutant and control strains (*Figure 4—figure supplement 1D*). Among mixtures exposed to 640 μM 4-HNE, unmarked WT and marked WT showed no significant difference in 4-HNE survival (*Figure 4A*). Loss of *rha2* had no effect on 4-HNE survival while Δ*rha1* had a modest fivefold reduction relative to WT. However, the Δ*rha1*Δ*rha2* mutant exhibited a 50-fold competitive defect compared to WT *L. monocytogenes* that was rescued by either *rha1* or *rha2* expression in trans, demonstrating that both genes must be absent for the toxic effect to manifest (*Figure 4A*). We also tested *L. monocytogenes* WT and Δ*rha1*Δ*rha2* survival in the presence of heat and diamide and found no significant difference between WT and the double mutant (*Figure 4B,C*).

We next assessed 4-HNE protein adduct accumulation by dot blot in WT, Δ*rha1*Δ*rha2*, Δ*rha1*Δ*rha2::rha1* and Δ*rha1*Δ*rha2::rha2* strains after 4-HNE exposure. We found that following 4-HNE treatment there was a modest but significant twofold increase in 4-HNE adduct accumulation

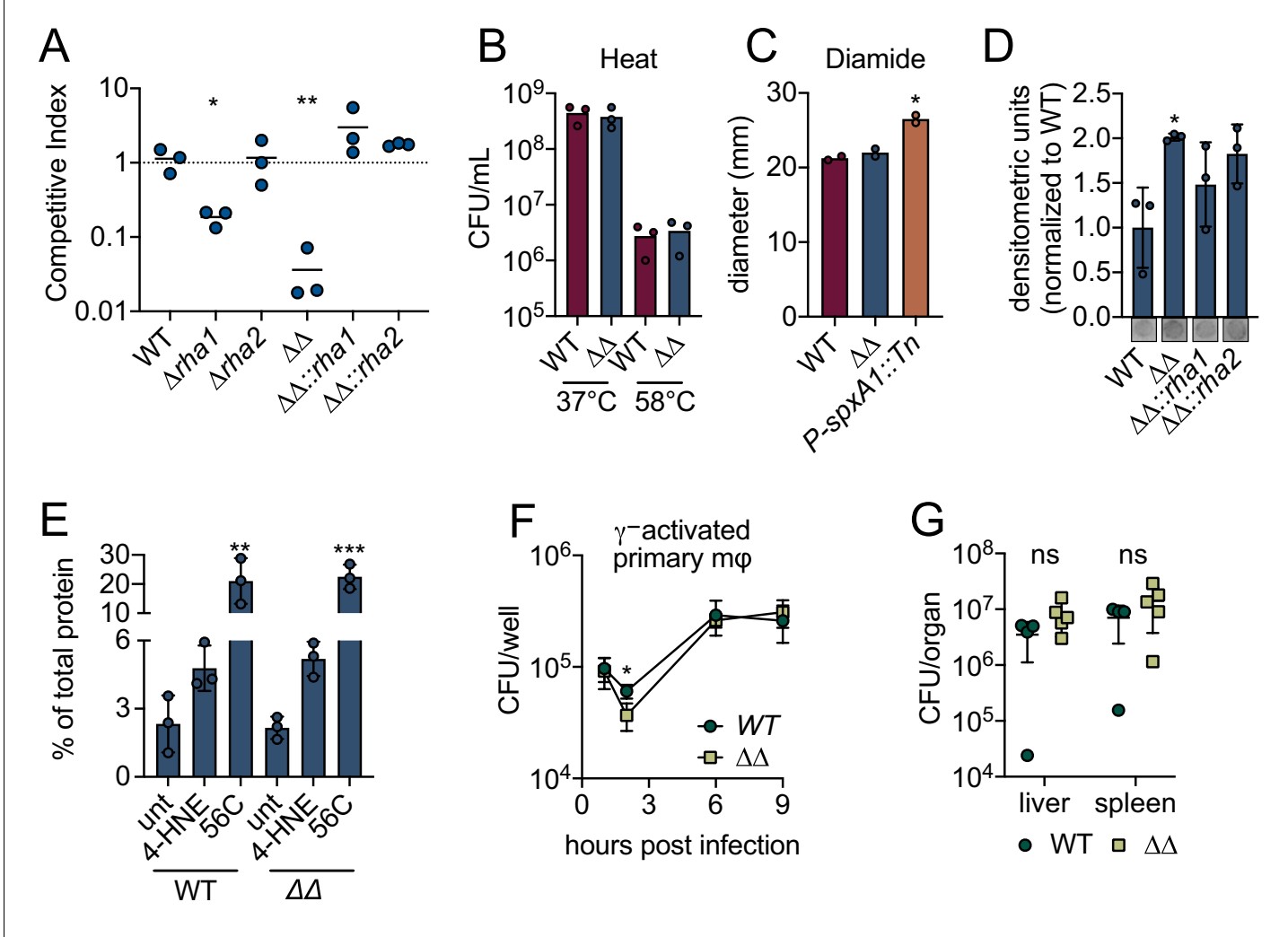

**Figure 4.** *L. monocytogenes* Δrha1Δrha2 has reduced ability to survive 4-HNE toxicity. (A) Competitive index of WT and mutant *L. monocytogenes* in PBS treated with 640 µM 4-HNE at 37°C for 1 hr. (B) CFU of WT and Δrha1Δrha2 *L. monocytogenes* in TSB exposed to 58°C or 37°C for 15 min. (C) Diameter of zone of clearance by 1M diamide on lawns of WT, Δrha1Δrha2 and positive control *P-spxA1::tn L. monocytogene*s on TSA plates as described in *Reniere et al., 2016*. (D) Accumulation of 4-HNE-adducted proteins in *L. monocytogenes* exposed to 640 µM 4-HNE for 3 hr in TSB media, assessed by dot blot and normalized to WT. Dot blot images below are representative. (E) Aggregated protein found in the insoluble fraction measured as percent of total protein in WT and Δrha1Δrha2 *L. monocytogenes*. Untreated, 4-HNE treatment (640 µL for an hour) and heat shock (56°C for 10 min). (F) CFU/well of WT and Δrha1Δrha2 *L. monocytogenes* in recombinant murine IFN-γ (100 ng) activated WT primary murine macrophages. (G) CFU/organ of WT and Δrha1Δrha2 *L. monocytogenes* at 48 hr intravenous murine infection. Data in figures (A) and (D) are in technical triplicate, representative of at least three independent experiments. Data in (C) are two independent experiments, with two pooled biological duplicates within each experiment. Data in (B), (E), (F) and (G) are biological triplicate. Statistics in (A) are unpaired t-tests between WT and mutant *L. monocytogenes* competition pairs. Statistics in (C) are unpaired t-tests between WT and mutant *L. monocytogenes*. Statistics in (D) and (E) are an ordinary one-way ANOVA with a Dunnett's multiple comparison test against WT (in D) or untreated sample (in E). Statistics in (F) are unpaired t-tests comparing WT and Δrha1Δrha2 *L. monocytogenes* CFU at hour two post infection. Statistics in (G) are unpaired t-tests comparing WT and Δrha1Δrha2 *L. monocytogenes* CFU within each organ. Error bars are mean ± SD. *p<0.05; **p<0.01; ***p<0.001. In figures (A) and (G), the line is drawn at the median of data. The online version of this article includes the following figure supplement(s) for figure 4:

**Figure supplement 1.** Impact of Δrha1 and Δrha2 on *L. monocytogenes* survival in PBS.

in the Δrha1Δrha2 mutant compared to WT (*Figure 4D*). Interestingly, although there was a modest reduction in adduct levels in both complement strains, expression of neither gene in trans fully restored WT levels of adduct formation, even though either *rha1* or *rha2* complementation fully restored bacterial survival in our competition experiment. Because 4-HNE-mediated protein cross-linking may preclude detection of 4-HNE adducts by dot blot, we subsequently assessed total

aggregation of proteins in WT *L. monocytogenes* and the Δ*rha1*Δ*rha2* mutant when exposed to 4-HNE by measuring protein content in the insoluble versus soluble fraction of cell lysates. We found that in WT *L. monocytogenes* 4-HNE does not lead to significant protein aggregation compared to the untreated control and significantly less insoluble protein accumulation compared to a 10-min exposure of 56°C (*Figure 4E*). In addition, the Δ*rha1*Δ*rha2* mutant did not exhibit an increase in protein aggregation after 4-HNE treatment compared to WT. Collectively, 4-HNE exposure had modest impacts on protein adduct formation and insoluble protein accumulation, supporting the conclusion that the resistance to the bactericidal effect of 4-HNE exposure conferred by Rha1/2 is independent of proteome damage.

Given the defect in the Δ*rha1*Δ*rha2* mutant viability compared to WT *L. monocytogenes* in vitro, we explored the impacts of these genes using tissue and murine infection models. During infection of IFN-γ activated primary murine macrophages, we observed no notable differences between the Δ*rha1*Δ*rha1* mutant and WT *L. monocytogenes* (*Figure 4F*). In mice infected via intravenous injection, no significant phenotype was observed at 48 hr post infection in either the spleen or the liver (*Figure 4G*). While Rha1 and Rha2 contribute to *L. monocytogenes* 4-HNE resistance in vitro, these genes are dispensable for this organism's capacity to counteract this metabolite in vivo.

## Recombinant Rha1 and Rha2 metabolize 4-HNE to 4-HNA

Our data suggested that Rha1 and Rha2 are expressed in response to 4-HNE and may contribute to *L. monocytogenes'* resistance to this compound. The predicted function of both Rha1 and Rha2 suggested they might act to directly metabolize 4-HNE. In order to determine if these putative reductases can utilize 4-HNE as a substrate, we generated recombinant Rha1 and Rha2 proteins. As controls for these studies, we generated catalytically dead variants of the two proteins by mutating amino acids predicted to be involved in flavin binding by Rha1 (asparagine-47) and NADPH binding by Rha2 (tyrosine-195) to alanine (*Figure 5A*). All proteins were expressed and characterized for NADPH oxidation in the presence and absence of 4-HNE (*Figure 5B*). As a positive control for NADPH-dependent 4-HNE turnover, we used the human Aldo-Keto Reductase 1C1 (AKR1C1), which metabolizes 4-HNE in a NADPH-dependent manner (*Burczynski et al., 2001*; *Figure 5C*). Only the WT variants of Rha1 and Rha2 exhibited NADPH oxidation upon addition of 4-HNE, consistent with their capacity to mediate NADPH-dependent reduction of the αβ-unsaturated aldehyde. We then measured NADPH oxidation of Rha1 and Rha2 using the aldehyde panel we previously used for our expression specificity analysis (*Figure 4—figure supplement 1A*; *Figure 3—figure supplement 2A*). We found that both Rha1 and Rha2 showed the most robust NADPH oxidation in the presence of 4-HNE, although Rha2 in particular showed modest NADPH oxidation with 4-HHE, perhaps suggesting a wider substrate range for Rha2 than Rha1 (*Figure 5D*).

Generally, NADPH-dependent 4-HNE reduction can occur at either the carbon-carbon double bond, generating the saturated aldehyde 4-hydroxynonanal (4-HNA), or on the carbonyl moiety, generating the alcohol 1,4-dihydroxynonene (1,4-DHN) (*Schaur et al., 2015*). To elucidate which of these two products Rha1 and Rha2 may be generating, we performed thin-layer chromatography (TLC) on their enzymatic products. We concurrently utilized the human enzyme AKR1C1 as a positive control for 1,4-DHN production (*Burczynski et al., 2001*) and the *Arabidopsis thaliana* enzyme P1-ZCr as the positive control for 4-HNA (*Mano et al., 2002*). We also chemically generated 1,4-DHN as an additional control through sodium borohydride reduction of 4-HNE. In reactions with either Rha1 or Rha2, a new spot was observed which required addition of NADPH and that co-migrated with the 4-HNA product formed by P1-ZCr (*Figure 5E*). These results support the conclusion that both Rha1 and Rha2 have the capability to directly metabolize 4-HNE to 4-HNA (*Figure 5F*).

## Ectopic expression of *rha1* and *rha2* confers 4-HNE resistance to the sensitive bacteria *B. subtilis*

Based on our recombinant protein data, we hypothesized that Rha1 and Rha2 could confer 4-HNE resistance to a sensitive organism. To this end, we utilized *B. subtilis* as the host for heterologous expression of *rha1* and *rha2*. *B. subtilis* is exquisitely sensitive to 4-HNE toxicity, exhibiting 300-fold reduction in recoverable CFU compared to *L. monocytogenes* upon 4-HNE exposure as well as a significant growth delay (*Figure 2A*, *Figure 6—figure supplement 1A*). We ectopically expressed both *rha1* and *rha2*, and their corresponding catalytically dead variants individually and in combination in

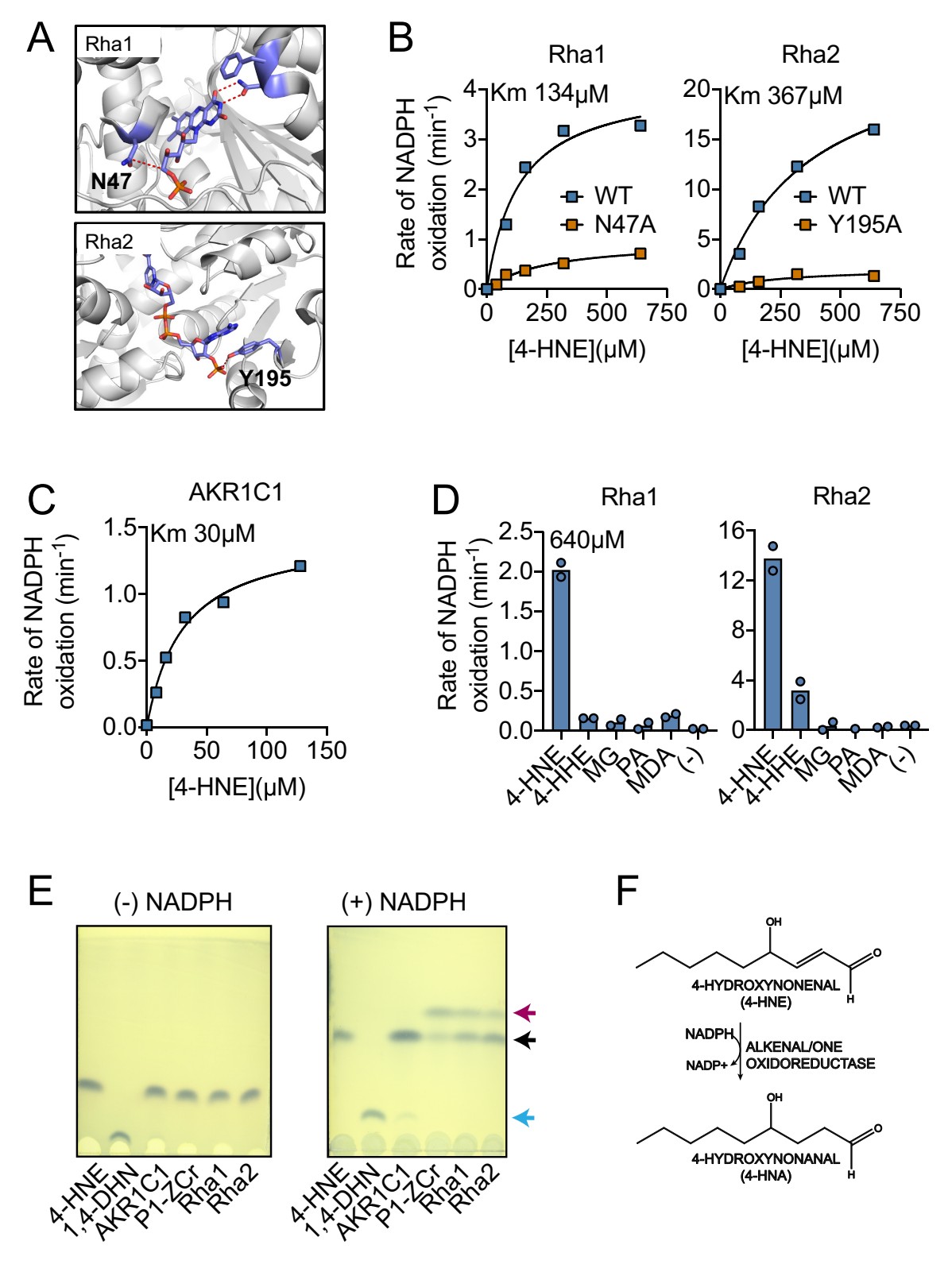

**Figure 5.** Recombinant Rha1 and Rha2 metabolize 4-HNE to 4-HNA. (**A**) Phyre2 structural homology models predict that asparagine-47 interacts with the FMN in Rha1 (left) and tyrosine-195 coordinates NADPH in Rha2 (right). (**B**) Rates of NADPH oxidation (200 µM) by WT (blue) and mutant (orange) variants of Rha1 and Rha2 in the presence of 4-HNE. (**C**) Rate of NADPH oxidation (200 µM) by human enzyme AKR1C1 in the presence of 4-HNE. (**D**) NADPH oxidation rate by Rha1 and Rha2 in the presence of 640 µM final concentration of various aldehydes. (**E**) TLC plates showing the migration of

*Figure 5 continued on next page*

*Figure 5 continued*

reaction contents of Lane 1: 4-HNE, Lane 2: 1,4-DHN, and Lanes 3–6: indicated enzymes with 4-HNE in the absence and presence of NADPH after 1 hr of reaction at room temperature. 4-HNE – black arrow, 4-HNA – red arrow, 1,4-DHN – blue arrow. (F) Diagram of 4-HNE to 4-HNA conversion. Data in (B), (C), (D) and (E) are representative of at least three independent experiments.

*B. subtilis* in the presence of 4-HNE. We compared the growth of *B. subtilis* expressing the active forms of the enzymes to their catalytically dead counterparts. Consistent with our observations of the *L. monocytogenes rha* deletion mutants, expression of *rha2* in *B. subtilis* had no effect on growth in the presence of 4-HNE under the conditions tested, *rha1* reproducibly reduced lag time by approximately 50 min in treatment with 640 µM 4-HNE (*Figure 6—figure supplement 1B*). Expression of both *rha1* and *rha2* had the largest growth rescue, reducing lag time by up to 3 hr (*Figure 6A*). Further characterization focused on *B. subtilis* expressing both *rha1* and *rha2* genes, as this strain had the most robust phenotype. When assessed for bacterial survival following 4-HNE treatment, *B. subtilis* expressing the functional enzymes exhibited nearly a 2-log survival advantage relative to the control strain expressing enzymatically dead *rha1/2* (*Figure 6B*). Additionally, soluble cellular fractions from *B. subtilis* exposed to 4-HNE and probed for 4-HNE protein adducts by dot blot revealed a ~ 70% reduction in 4-HNE conjugates in the *B. subtilis* strain expressing both of the active *rha1* and *rha2* genes versus their catalytically dead counterparts (*Figure 6C*, *Figure 6—figure supplement 1C*).

To determine if 4-HNE resistance conferred by expressing *rha1/2* could contribute to bacterial survival within mammalian cells, *B. subtilis rha1/2* strains were assessed for viability following phagocytosis by primary bone-marrow-derived macrophages. We found that *B. subtilis* expressing the active forms of Rha1 and Rha2 (WT) maintained a significantly higher CFU over the course of 8 hr than the *B. subtilis* expressing the catalytically dead forms (MUT) (*Figure 6D*). To determine whether this survival advantage was due to 4-HNE resistance, we measured *B. subtilis* survival within bone marrow-derived macrophages from *gp91^{phox-/-}* mice deficient in oxidase cytochrome b-245, which are unable to produce the reactive oxygen burst and therefore 4-HNE (*Esterbauer et al., 1991*). Consistent with the role of *rha1* and *rha2* in mediating resistance to a ROS-derived factor, the protective effect of WT *rha1/2* expression was eliminated in the absence of *gp91^{phox}* (*Figure 6D*). Together, these observations revealed that expression of *rha1* and *rha2* in *B. subtilis* imparts resistance to 4-HNE toxicity and impacts bacterial survival in response to the host cell's ROS burst.

## Discussion

In this study, we provide evidence that the ROS-derived metabolite 4-HNE accumulates during *L. monocytogenes* infection in both tissue culture and in a murine model of infection. We also show that 4-HNE exhibits antimicrobial effects in several bacterial species and that in the highly resistant intracellular pathogen *L. monocytogenes*, exposure to this aldehyde induces a broad and robust transcriptional profile. Among the highest induced genes are components of the heat shock response, consistent with aldehyde induced protein damage, a known effect of 4-HNE exposure. In addition, two genes, *rha1* and *rha2,* are highly and specifically induced by 4-HNE exposure, and these two enzymes reduce 4-HNE to 4-HNA in an NADPH-dependent manner in vitro. Disruption of *rha1* and *rha2* in *L. monocytogenes* results in a decrease in viability in the presence of 4-HNE in vitro but not in vivo, and heterologous expression of *rha1* and *rha2* in *B. subtilis*, a non-pathogenic 4-HNE-sensitive organism, conferred increased tolerance to 4-HNE toxicity. Rha1 and Rha2 expression in *B. subtilis* also allowed for greater survival following phagocytosis by bone-marrow-derived macrophages in a manner entirely dependent upon phagocyte ROS generation. Together this work supports the conclusion that 4-HNE represents one of the individual molecular components of ROS-mediated host defense through its direct antimicrobial effects on bacteria and that pathogens have likely evolved complex mechanisms of surviving its encounter within eukaryotic hosts.

There are many parallels between the chemical and biological functions of 4-HNE and other toxic metabolites that function in antimicrobial defense. The freely diffusible and highly reactive diatomic gas nitric oxide (NO) is produced during infection (*Iyengar et al., 1987*; *Stuehr and Marletta, 1985*) and has a direct role in preventing bacterial growth (*Nathan and Hibbs, 1991*). However, due to the conservation of its reactive targets, elevated levels of NO also exert pathological effects

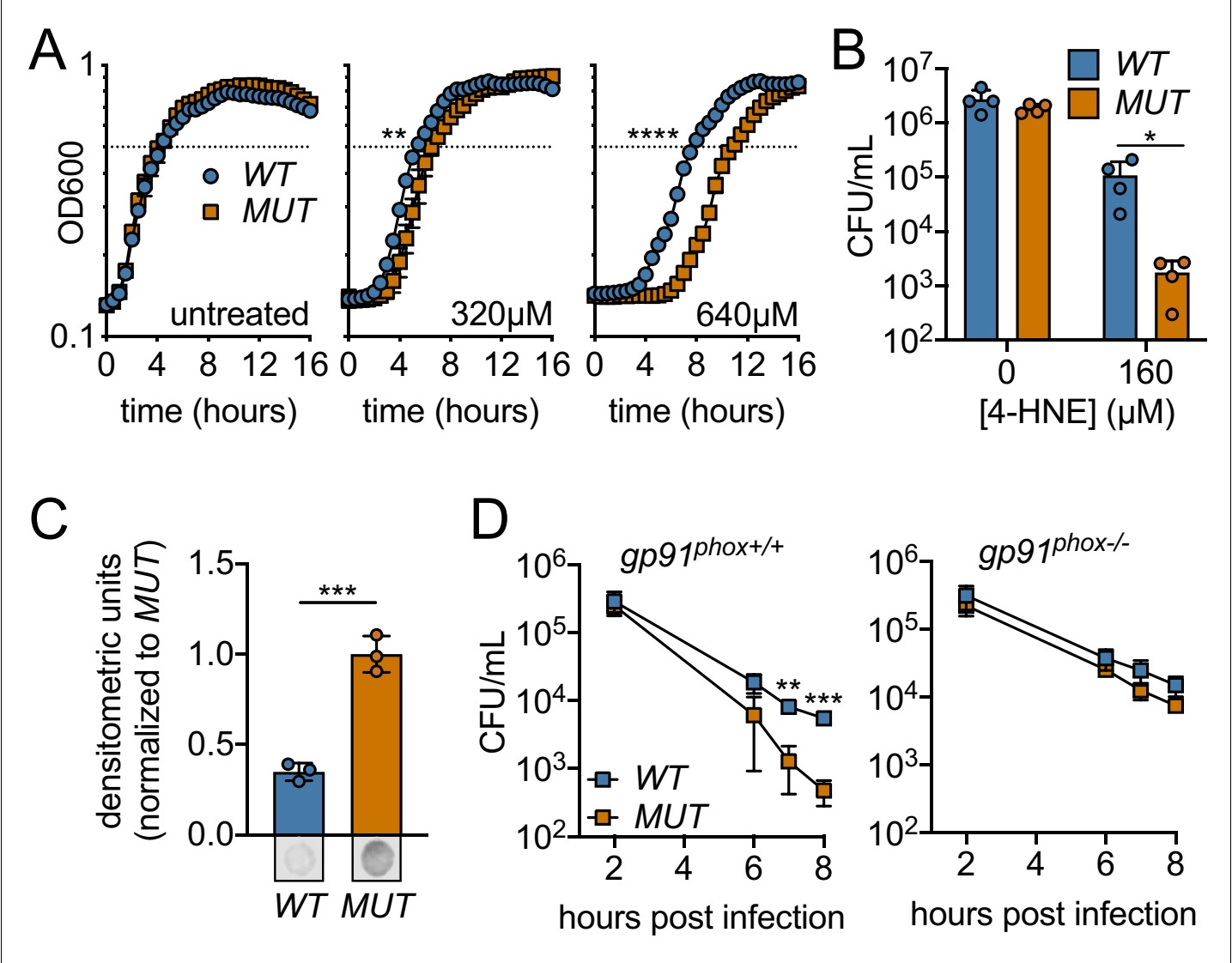

**Figure 6.** Ectopic expression or Rha1 and Rha2 confers 4-HNE resistance to sensitive bacteria. (**A**) Growth of *B. subtilis* expressing either the *WT* or *MUT* (catalytically dead mutant) versions of *rha1* and *rha2* in TSB at 37°C in the presence of the indicated concentrations of 4-HNE added at time zero. Dotted line represents $OD_{600}$ 0.5. (**B**) Survival of *B. subtilis* WT and MUT in PBS at 37°C with 160 μM 4-HNE for 1 hr. (**C**) 4-HNE conjugates from *B. subtilis* WT and MUT soluble cell lysates (3 μg total protein) 3 hr after 4-HNE treatment as assessed by dot blot and quantified by densitometry on a Licor Odyssey Fc. Dot blots below are representative. (**D**) *B. subtilis* WT and MUT survival following phagocytosis by Interferon gamma-activated primary WT or phagosomal oxidase-deficient bone-marrow-derived macrophages (WT or *gp91^phox-/-* pBMMs). All experiments were performed in biological triplicate. Statistics in (**A**) are unpaired t-tests comparing the hours to $OD_{600}$ 0.5 between *WT* and *MUT B. subtilis pHT01::rha1/2*. Statistics in (**B**), (**C**) and (**D**) are unpaired t-tests comparing *WT* and *MUT B. subtilis pHT01::rha1/2*. Error bars are mean ± SD. *p<0.05; **p<0.01; ***p<0.001; ****p<0.0001.

The online version of this article includes the following figure supplement(s) for figure 6:

**Figure supplement 1.** Impacts of 4-HNE, Rha1, and Rha2 on 4-HNE growth delay and adduct accumulation in a sensitive bacteria.

during both sterile inflammation and acute infections (*Nagafuji et al., 1995*; *Galley and Webster, 1998*). 4-HNE is membrane diffusible, highly reactive, and contributes to disease pathology due to its cytotoxic activity toward eukaryotic cells. These parallels, together with our findings that bacterial infection induces 4-HNE production are consistent with the premise that 4-HNE represents a component of ROS-mediated host defense, among such other toxic metabolites as superoxide, hydrogen peroxide, and hypochlorite.

While a role for 4-HNE in host antimicrobial defense has yet to be appreciated in mammals, plants utilize a variety of lipophilic molecules generated by the oxidation of polyunsaturated fatty acids (PUFAs), collectively referred to as oxylipins. While generally considered to be involved in signal transduction, many oxylipins can directly inhibit bacterial growth (*Prost et al., 2005*) and 4-HNE itself is a component of the oxylipin burst in soybean where it serves an anti-fungal function (*Vaughn and Gardner, 1993*). Because 4-HNE is one of several distinct metabolites produced following oxidation of PUFAs in mammals, it is conceivable that other reactive byproducts of this process also contribute to microbial defense in a similar manner.

The contrasting observations between the liver and spleen were somewhat surprising given our observation that infection of murine hepatocytes in tissue culture with *L. monocytogenes* in vitro induced accumulation of 4-HNE conjugates. These observations are likely a consequence of several factors. The liver is a major site of small molecule detoxification and hepatocytes are known to produce high levels of many of the 4-HNE metabolizing proteins, including aldo-keto reductases, alcohol dehydrogenase, and the 4-HNE glutathione transferase, GSTA4 (*Zheng et al., 2014*). Additionally, the immune-driven mobilization of arachidonic acid and ROS precursors that lead to 4-HNE generation may result in elevated accumulation of this host aldehyde in the spleen. These observations suggest that the need to counteract 4-HNE toxicity may be distinct depending upon the tissue tropism of an infecting pathogen.

To survive within the sterile tissues of eukaryotic hosts, bacterial pathogens often counteract the toxic effects of the immune response. Our discovery of two genes that confer synergistic resistance to 4-HNE in *L. monocytogenes* begin to provide insight into the mechanisms by which 4-HNE toxicity might be overcome. In vitro studies suggest that Rha1 and Rha2 both metabolize 4-HNE in an NADPH-dependent manner to 4-HNA, suggesting redundant functions. Redundancy in bacterial resistance to ROS is a relatively common phenomenon, including the need to eliminate five individual enzymes in *Salmonella enterica* Serovar Typhimurium to exhibit a phenotype in the presence of hydrogen peroxide (*Hébrard et al., 2009*) and simultaneous disruption of four enzymes in *Bacillus anthracis* to observe a phenotype in the presence of superoxide (*Cybulski et al., 2009*). Such redundancy in bacterial detoxification programs likely decreases the chances that genetic drift or other genomic damage would render an organism defenseless against oxidative stress. It remains unclear, however, why both Rha1 and Rha2, which have similar in vitro enzymatic properties, are both required for 4-HNE resistance following exposure to the pure compound. If their effects were simply redundant, the loss or addition of both genes in *L. monocytogenes* and *B. subtilis*, respectively, would be additive. Our findings are contrary to this expectation, suggesting a more complex role. We speculate that either alternative localization of these proteins resulting in detoxification within specific subcellular compartments or unidentified alternative roles in mediating 4-HNE resistance are at play.

A wide range of susceptibility to 4-HNE exposure was observed among the various bacterial species tested in this study. These observations may reflect unique mechanisms by which these organisms combat 4-HNE as well as potential conservation of the Rha1 and Rha2 proteins. Among the most 4-HNE-sensitive organisms, *B. subtilis*, which resides in the soil, likely does not encounter 4-HNE and appears to have no resistance to its exposure, while another highly 4-HNE-sensitive bacterium, *F. novicida,* is known to block the generation of ROS by the host NADPH oxidase, perhaps limiting the need to detoxify this metabolite directly (*Mohapatra et al., 2010*). *E. faecalis* and *L. monocytogenes* were the most resistant organisms tested. Among all organisms tested, *E. faecalis* has the clearest homologs based on sequence identity to Rha1 (59%) and Rha2 (71%) from *L. monocytogenes*. It is difficult to predict substrate specificity of flavin-dependent and NADPH-dependent reductases solely on sequence conservation, and without direct enzymological characterization, homologous function cannot be concluded. Additionally, while these two genes are most robustly induced by *L. monocytogenes* 4-HNE exposure, it cannot be concluded that they are not a part of a broader stress response. In particular, Rha2 is induced by diamide stress and exhibits enzymatic activity toward 4-HHE, suggesting a wider role in stress responses and substrate promiscuity. Given that many flavin and NADPH-dependent reductases have roles in detoxifying endogenous enone containing compounds, like quinones, or other exogenous electrophilic toxic metabolites susceptible to Michael-addition, including nitroaromatic compounds, it is certainly feasible that these enzymes play roles beyond 4-HNE resistance.

While Rha1 and Rha2 both contribute to 4-HNE resistance, the Δrha1Δrha2 L. monocytogenes strain still exhibits several logs of survival benefit relative to the related organism B. subtilis, suggesting that other mechanisms of 4-HNE resistance remain to be identified. Among the many uncharacterized genes induced during 4-HNE exposure, lmo0796 shows homology to bcnA, a secreted lipocalin in Burkholderia cenocepacia which sequesters long-chain lipophilic antibiotics (El-Halfawy et al., 2017). 4-HNE, with its long hydrophobic tail, could conceivably be neutralized in an analogous manner. It is also possible that many intrinsic resistance properties of L. monocytogenes are not reflected through transcriptional responses. For instance, addition of amine containing constituents on the cell's surface through lysinylation of teichoic acids and/or lipids, as well as deacetylation of peptidoglycan, may provide a nucleophile reactivity barrier that prevents 4-HNE entry into the bacterial cell. Additionally, αβ-unsaturated aldehydes have preferential reactivity toward sulfhydryl groups, including cysteine and glutathione, and it is expected that thiolate depletion would be the major mechanism of 4-HNE toxicity (LoPachin and Gavin, 2014). Indeed, the thiol responsive transcription factor spxA1 was induced >2 fold in response to 4-HNE and the magnitude of heat shock gene induction mirrored results reported for B. subtilis following diamide treatment, a potent inducer of disulfide stress (Leichert et al., 2003). While our findings provide initial molecular insight into one pathogen's resistance to 4-HNE, it is clear that many details are yet to be revealed.

Taken together, our findings extend the range of antimicrobial molecules generated through the reactive oxygen burst to include the byproducts of lipid peroxidation. Additionally, bacteria whose infection cycles involve intimate exposure to these molecules, such as L. monocytogenes, have the capacity to resist this toxicity. Future investigation of the impacts of 4-HNE on a diverse array of organisms with varied infection models will highlight the importance of this metabolite on host defense and the varied mechanisms by which pathogens counteract its toxicity to promote infection.

## Materials and methods

**Key resources table**

| Reagent type (species) or resource | Designation | Source or reference | Identifiers | Additional information |
| --- | --- | --- | --- | --- |
| Cell line (Mus musculus) | J774A.1 | PMID:1612739 | | Immortalized murine macrophages |
| Cell line (Mus musculus) | TIB73 | ATCC: BNL CL.2 | | Immortalized murine hepatocytes |
| Strain, strain background (Mus musculus) | C57BL/6J | Jackson Laboratories: 000664 | | |
| Strain, strain background (Mus musculus) | gp91phox- (C57BL/6J backcross) | Jackson Laboratories: 002365 | | |
| Strain, strain background (Escherichia coli) | DH10b | Invitrogen: 10536193 | | |
| Strain, strain background (Escherichia coli) | SM10 | DOI: https://doi.org/10.1038/nbt1183-784 | | |
| Strain, strain background (Escherichia coli) | BL21 (DE3) | EMD Millipore: 69450 | | |
| Strain, strain background (Escherichia coli) | Rosetta (DE3) | EMD Millipore: 70954 | | |
| Strain, strain background (Listeria monocytogenes) | 10403S | PMID:2125302 | WT | |
| Strain, strain background (Bacillus subtilis) | 168 | PMID:18723616 | WT | |
| Strain, strain background (Francisella novicida) | U112 | PMID:17550600 | WT | |
| Strain, strain background (Pseudomonas aeruginosa) | PAO1 | PMID:20023018 | WT | |

*Continued*

| Reagent type (species) or resource | Designation | Source or reference | Identifiers | Additional information |
|---|---|---|---|---|
| Strain, strain background (*Staphylococcus aureus*) | Newman | PMID:17951380 | WT | |
| Strain, strain background (*Enterococcus faecalis*) | OG1RF | PMID:18611278 | WT | |
| Strain, strain background (*Listeria monocytogenes*) | DP-L3903 | PMID:11500481 | Antibiotic marked competition strain | |
| Strain, strain background (*Listeria monocytogenes*) | DP-L4056 | PMID:12107135 | Phage-cured 10403S strain for use with pPL1 integration plasmid | |
| Genetic reagent | Genetically modified bacterial strains used in this work | | This paper | *Supplementary file 1* |
| recombinant DNA reagent | pPL1077 | This study | pPL1-GFP integration plasmid | (Gift from Peter Lauer; Berkeley, CA) |
| Recombinant DNA reagent | pKSV7 | PMID:8388529 | Shuttle vector for *L. monocytogenes* gene disruption | |
| Recombinant DNA reagent | pET20b | Novagen: 69739 | C-terminus 6xHis tagged *E. coli* T7 expression vector | |
| Recombinant DNA reagent | pET28b | Novagen: 69865 | N-terminus 6xHis tagged *E. coli* T7 expression vector | |
| Recombinant DNA reagent | pPL2-Pspac | PMID:25583510 | Integrative *L. monocytogenes* plasmid pPL2 engineered with constitutive Pspac promoter | |
| Recombinant DNA reagent | pHT01 | PMID:17624574 | *B. subtilis* expression vector | |
| Recombinant DNA reagents | Plasmids generated and used in this work | | This paper | *Supplementary file 2* |
| Gene (*Listeria monocytogenes*) | *lmo0103 (LMRG_02352)* | GenBank: Gene ID 12552319 | Gene *rha1* | |
| Gene (*Listeria monocytogenes*) | *lmo0613 (LMRG_00296)* | GenBank: Gene ID 12552833 | Gene *rha2* | |
| Gene (*Homo sapiens*) | *akr1c1* | GenBank: Gene ID 1645 | Gene *akr1c1* | |
| Gene (*Arabidopsis thaliana*) | *p1-zcr* | GenBank: Gene ID 831560 | Gene *p1-zcr* | |
| Sequence-based reagents | RT-qPCR primers | This paper | | (See *supplementary file 3*) |
| Sequence-based reagents | Cloning primers | This paper | | (See *supplementary file 3*) |
| Sequence-based reagents | Gene coding sequences for protein expression | This paper | | (See *supplementary file 4*) |
| Chemical, compound, drug | 4-HNE | Cayman Chemical: 32100 | 64 mM 4-HNE in absolute ethanol | |
| Antibody | Anti-4-HNE adduct | Abcam: ab46545 | Rabbit IgG polyclonal antibody | (1:200) |
| Antibody | Anti-actin | Abcam: ab8226 | Mouse IgG monoclonal antibody | (1:1000) |
| Antibody | Anti-rabbit secondary antibody | Licor: 926–32211 | Goat IgG monoclonal antibody | (1:8000) |
| Antibody | Anti-mouse secondary antibody | Licor: 926–68072 | Donkey IgG monoclonal antibody | (1:8000) |
| Antibody | Anti-GFP primary antibody | Invitrogen: MA5-15256 | Mouse IgG monoclonal antibody | (1:500) |

*Continued on next page*

*Continued*

| Reagent type (species) or resource | Designation | Source or reference | Identifiers | Additional information |
|---|---|---|---|---|
| Antibody | Isotype control primary antibody, histology | R and D Systems: AB-105-C | Normal Rabbit IgG | (1:1000) |
| Commercial assay, kit | Ovation Complete Prokaryotic RNA-Seq System | Nugen: 0363–32 | RNA-seq processing kit | |
| Software, algorithm | Rockhopper | cs.wellesley.edu/~btjaden/Rockhopper/ | RNA-seq data analysis software | |

## Statistics and reproducibility

Sample sizes were defined as at least n = 3 for all experiments unless otherwise noted. Biological replicates are defined as bacterial samples grown from independent colonies. Technical replicates are defined as bacterial samples grown from the same colony and split for treatment and processing. Independently performed experiments are defined as being done on different days. Type of replication, number of replicates, statistical tests performed, and definitions of significance symbols are indicated in figure legends. All statistics were performed using Prism Version 8.4.2 Software. Data points represent mean ± SD of replicate experiments. Statistical outliers were not excluded in this study. For murine infection studies a group size of five mice was selected. This was based upon the ability to detect a 1-log effect on bacterial burdens between groups, with a 45% standard deviation on log transformed CFU measures, an alpha of 0.05 and power of 0.9. These parameters were selected based upon previous *L. monocytogenes* infection studies performed in the laboratory.

## Cell lines

TIB73 cells were purchased from ATCC. J774A.1 cells were obtained from the laboratory of Dr. Michelle Reniere and were confirmed via STR profiling by ATCC. Both cell lines were tested negative for Mycoplasma contamination.

## Strains and routine growth conditions

Unless otherwise specified, *L. monocytogenes* was grown shaking at 37°C degrees in tryptic soy broth (TSB) media and *E. coli* at shaking at 37°C degrees in Luria-Bertani (LB) media with appropriate antibiotic selection. Unless otherwise noted, *B. subtilis* was struck on LB plates with appropriate antibiotics and induction agent overnight at 30°C, after which the biomass was scraped off the plates, resuspended in LB media, passed 6 to 10 times through a 27-gauge needle to break up clumps and chains, then normalized to an $OD_{600}$ of 1. When required for selection, antibiotic concentrations used in this study were as follows – *L. monocytogenes* selections: streptomycin 200 µg/mL, chloramphenicol 5 µg/mL; *E. coli* selections: ampicillin 50 µg/mL; *B. subtilis* selections: chloramphenicol 10 µg/mL; tissue culture: gentamicin 50 µg/mL.

## DNA manipulation and plasmid construction

All DNA manipulation procedures followed standard molecular biology protocols. Primers were synthesized and purified by Integrated DNA Technologies (IDT). HiFi polymerase (Kapa Biosystems, #KK2102), FastDigest restriction enzymes (Thermo Fisher Scientific #FD0274), and T4 DNA ligase (Thermo Scientific # K1423) were used for plasmid construction, with the exception of *pHT01::rha1/2_WT* and *pHT01::rha1/2_MUT* which were generated using Gibson Assembly MasterMix (NEB, #E2611S). DNA sequencing was performed by Genewiz Incorporated.

## Bacterial infection and exogenous 4-HNE TIB73 dot blot

TIB73 cells were infected with *L. monocytogenes* following a previously developed protocol (*McFarland et al., 2017*). Exogenous 4-HNE (Cayman Chemical, #32100) was added to uninfected TIB73 cells by first washing the cells with sterile PBS and then adding 2 mL sterile PBS containing a final concentration of 10 µM 4-HNE for 10 min. TIB73 cells were lysed in whole cell lysis buffer (50 mM Tris pH 7.5, 150 mM NaCl, 1% Triton X-100) with EDTA (1 µM) and Halt Protease Inhibitor

Cocktail (Thermo Fisher Scientific, #78442). Protein concentration was determined using the Pierce BCA protein assay kit (Fisher Scientific, #PI23227). Lysates were resuspended in 1X PBS to achieve 2 µg of protein per 3 µl, which was the volume spotted out onto nitrocellulose membrane (Bio-Rad, #1620115). The nitrocellulose was then dried, blocked for 45 min in 5% dry milk, washed three times with TBS-T (Tris-buffered saline with 0.1% Triton X-100) and primary 4-HNE antibody was added at 1:200 dilution (Abcam, #ab46545). The antibody was incubated overnight at 4°C with rocking. Primary actin antibody (Abcam, #ab8226) was added at 1:1000 for 3 hr at room temperature. The primary antibodies were then washed off with TBS-T three times and secondary antibodies (Licor, #926–32211, #926–68072) were added at 1:8000 for 45 min at RT. The secondary antibodies were then washed with TBS-T twice, then TBS once and the blot was imaged on a Licor Odyssey Fc (Li-Cor, Inc). Relative densitometric analysis was performed using Licor Image Studio software.

## Mouse infections

*L. monocytogenes* was grown overnight statically at 30°C in Brain Heart Infusion (BHI) broth, then back-diluted using 1.2 mL of overnight culture to 4.8 mL of fresh BHI and grown for 1 hr at 37°C shaking. $OD_{600}$ of these cultures were taken and, using the conversion of 1 $OD_{600}$ = $1.7 \times 10^9$ CFU, diluted to $5 \times 10^5$ CFU/mL with PBS. 200 µl were then injected into female WT C57BL/6 mice between 6 and 8 weeks of age retro-orbitally ($1 \times 10^5$ CFU/mouse) and livers and spleens were harvested at 48 hr post infection. Livers were homogenized in 10 mL of cold 0.1% IGEPAL and spleens were homogenized in 5 mL using a Tissue Tearor Model 985370 (Biospec Products) at 10,000 RPM for 5 s/organ. Homogenates were diluted in PBS and plated on LB plates to enumerate CFU. All protocols were reviewed and approved by the Institutional Animal Care and Use Committee at the University of Washington.

## 4-HNE histology

Two female WT C57BL/6 mice were infected as outlined above, in addition to one uninfected control mouse. The livers and spleens were harvested at 48 hr post infection and placed in 10% neutral buffered formalin for 24 hr, after which the organs were removed from formalin and placed in PBS for 24 hr. Paraffin-embedded tissues were sliced and prepared as slides. Slides were then deparaffinized for 30 min at 60°C. All subsequent manipulations were performed on a Leica Bond Automated Immunostainer. Antigen retrieval for GFP was performed by HIER 2 (EDTA) treatment for 20 min at 100°C. Antigen retrieval for 4-HNE was performed by citrate treatment for 20 min at 100°C. Then a Leica Bond peroxide block was performed for 5 min at room temperature, and normal goat serum (10% in TBS) was added for 20 min at room temperature. Primary antibody was added (GFP 1:500; Rabbit IgG 1:1000; 4-HNE 1:200) (Invitrogen: #MA5-15256; R and D Systems: #AB-105-C; Abcam: #ab46545) in Leica Primary antibody diluent (Leica: #AR9352), for 30 min at room temperature. Leica Bond Polymer was added for 8 min at room temperature, after which the samples were washed with Leica Bond Mixed Refine (DAB) (Leica: # DS9800) detection solution twice for 10 min at room temperature. Hematoxylin Counterstain was added for 4 min and the samples were cleared to xylene. Finally, samples were mounted with synthetic resin mounting medium on a 1.5 cm coverslip and imaged with a Hamamatsu Nanozoomer Whole Slide Scanner and a Keyence BZ-X710 Microscope.

## *L. monocytogenes* PBS 4-HNE dot blots

*L. monocytogenes* were sub-cultured from overnight stationary phase cultures 1:100 into fresh media and grown to mid-log (0.4–0.8 $OD_{600}$). The bacteria were normalized to $OD_{600}$ 1, washed twice and resuspended in sterile PBS. A range of 4-HNE concentrations were added to the bacteria and the samples were placed at 37°C for 30 min. Upon completion, the bacteria were washed twice with PBS and spun at 10,000 x g for 5 min, then resuspended in fresh PBS. The bacteria were then sonicated using a narrow tip sonicator at 20% power, 1 s on 1 s off for 10 s and placed on ice. The bacteria were then spun at 4°C at 10,000 x g for 30 min. The subsequent lysate was transferred to fresh Eppendorf tubes containing Halt Proteinase and Phosphatase Inhibitor (Thermo Fisher Scientific, #78442) and stored at −80°C until use. For dot blots, the protein concentration was normalized using BCA (Fisher Scientific, #PI23227) and 3 µg in 3 µL was spotted onto nitrocellulose membrane (Bio-Rad, #1620115). The nitrocellulose was then dried, blocked for 45 min in 5% dry milk, washed three times with TBS-T and primary 4-HNE antibody was added at 1:200 dilution (Abcam,

#ab46545). The antibody incubated overnight at 4°C with rocking. The primary antibody was then washed off with TBS-T three times and secondary antibody (Licor, #926–32211) was added at 1:8000 for 45 min at room temperature. The secondary antibody was then washed off with TBS-T twice, TBS once and the blot was imaged on a Licor Odyssey Fc. Relative densitometric analysis was performed using Licor Image Studio software.

## RNA extraction from broth cultures of *L. monocytogenes*

*L. monocytogenes* were sub-cultured from overnight stationary phase cultures 1:100 into fresh media and grown to mid-log (0.4–0.8 $OD_{600}$). Then a final concentration of 640 µM 4-HNE (Cayman #32100) or vehicle (100% ethanol) was added to the bacteria, which continued to grow at 37°C shaking for 20 min. After 20 min, ice cold 100% methanol was added in equal volume to the culture flask and placed at −20°C overnight. The next day the bacteria were spun down and resuspended in 400 µL AE buffer (50 mM NaOAc pH 5.2, 10 mM EDTA in molecular grade water). The resuspended bacteria were then mixed with 400 µL acidified 1:1 phenol:chloroform pH 5.2 (Fisher Scientific, # BP1753I) and 40 µL 10% sodium dodecyl sulfate (SDS) and was vortexed for 10 min in a multi-tube vortexer. The tubes were then transferred to a 65°C heat block for 10 min, after which the mixture was transferred to a Heavy Phase-lock tube (VWR #10847–802) and spun down for 5 min at 17,000 x g. Then the aqueous layer was transferred into tubes containing 1 mL 100% ethanol and 40 µL 3M NaOAc and placed at −20°C for 6 hr. Then tubes were spun at 17,000 x g for 30 min at 4°C, the ethanol was aspirated and 500 µL 70% ethanol was added. The tubes were then centrifuged at 17,000 x g for 10 min at room temperature and the supernatant was aspirated. The RNA pellet was then dried in a speed vacuum concentrator for 5 min and resuspended in RNA-free molecular grade water. The extracted RNA was then treated with DNase (Ambion Life Technologies #AM1907) for an hour at 37°C and used for downstream processing.

## RNA-sequencing

RNA was processed using the Ovation Complete Prokaryotic RNA-Seq Library System (NuGEN, #0363–32, 0326–32, 0327–32) according to the manufacturer's instructions to a final pooled library concentration of 3 nM. Libraries were sequenced on an Illumina HiSeq 2500 (SR50) at The Genomics Resource at the Fred Hutchinson Cancer Research Center. Image analysis and base calling were performed using Illumina's Real Time Analysis v1.18.66.3 software, followed by 'demultiplexing' of indexed reads and generation of FASTQ files using Illumina's bcl2fastq Conversion Software v1.8.4. Reads determined by the RTA software to pass Illumina's default quality filters were concatenated for further analysis. The FASTQ files were aligned and analyzed using Rockhopper software (*McClure et al., 2013*). These data have been deposited to the GEO and are accessible using accession number GSE150188.

## RT-qPCR

Bacteria were grown in the same manner as for RNA-seq except they were treated with either (a) 500 µM of each tested aldehyde (b) 5 mM diamide (c) heat (50°C) or (d) nitric oxide (1 mM of the NO donor DEA/NO) for 20 min in TSB media culture at mid-log (0.4–0.8 $OD_{600}$). The RNA was extracted by the acidified phenol method as listed above and DNase treated and reverse-transcribed using the iScript Reverse Transcription Supermix (Bio-Rad, #1708840). SYBR Green (Thermo Fisher Scientific # K0223) was then used to amplify genes of interest and CT values and relative expression were normalized using CFX Maestro Software (Bio-Rad #12004110).

## Intracellular RNA extraction

RNA extraction from macrophages was performed as previously described (*Sigal et al., 2016*). J774 macrophages were seeded at a density of 2.0 × $10^7$ cells/dish in three 150 mm dishes in 30 mL media and incubated overnight. The next day, overnight *L. monocytogenes* culture grown at 30°C was washed twice with PBS and added to the cells at a MOI of 50. After 1 hr, the cells were washed twice with PBS and media containing gentamicin was added. Eight hours post-infection, cells were washed once with PBS and lysed by addition of cold nuclease-free water. Lysate was collected by scraping and centrifugation at 800 x g for 3 min at 4°C. Supernatants were passed through 0.45 µm filters in a vacuum apparatus, and filters were collected in conical tubes. Filters were vortexed with

650 µL sterile AE buffer for 1 min and centrifuged briefly. Bacteria-containing AE buffer was collected and used for immediate RNA extraction as described above.

## 4-HNE survival assays bacterial panel

Bacteria (*Listeria monocytogenes*, *Enterococcus faecalis*, *Pseudomonas aeruginosa*, *Escherichia coli*, *Staphylococcus aureus*, *Bacillus subtilis*, and *Francisella novicida*) were inoculated overnight in TSBC (TSB + cysteine 0.1% required for *F. novicida* growth) and grown at 37°C. The next day the bacteria were sub-cultured 1:1000 into fresh TSBC and allowed to reach mid-log (0.4–0.8 $OD_{600}$). At mid-log, the ODs of the bacteria were normalized to $OD_{600}$ 1, washed twice in sterile PBS and resuspended in sterile PBS. Then the bacteria were diluted 1:100 into sterile PBS and various concentrations of 4-HNE were added. The bacteria were then placed at 37°C for an hour, then plated on TSAC (TSA + cysteine 0.1%) plates. Colonies were enumerated after overnight growth at 37°C.

## *L. monocytogenes* competition experiments

Colonies of *L. monocytogenes* were picked off BHI plates and inoculated into 2 mL TSB which were then grown shaking at 37°C to mid-log (0.4–0.8 $OD_{600}$). At mid-log the bacteria were normalized to $OD_{600}$ 1, washed twice in sterile PBS and resuspended in sterile PBS. Then the bacteria were diluted 1:100 into sterile PBS and appropriate strains were mixed together in a 1:1 ratio, after which 640 µM of 4-HNE or vehicle (ethanol) was added. The bacteria were then placed at 37°C for 1 hr. 2x concentrated TSB media was added to the bacterial-PBS solution and the cells recovered for an hour at 37°C. The bacteria were then plated on BHI-streptomycin and BHI-chloramphenicol (five plates for competitive strain differentiation. CFUs were enumerated after 24–48 hr of growth at 37°C). In competition between WT and WT, Δ*rha1*, Δ*rha2* and Δ*rha1*Δ*rha2*, the WT competition strain was the marked strain DPL-3903 (*Auerbuch et al., 2001*) In competition between WT and ΔΔ::*rha1* and ΔΔ::*rha2*, WT *L. monocytogenes* were unmarked while the complemented strains carried resistance to chloramphenicol.

## *L. monocytogenes* heat survival assay

Single colonies were inoculated into BHI-streptomycin and grown overnight at 37°C with shaking. Cultures were back diluted to $OD_{600}$ 0.03 in 25 mL fresh TSB- streptomycin in 125 mL flasks and grown with shaking at 37°C. At mid-log ($OD_{600}$0.3–0.6) the bacteria were shifted to static growth at either 58°C or 37°C for 15 min. The bacteria were incubated at 30°C for CFU on BHI plates overnight.

## *L. monocytogenes* diamide sensitivity assay

The protocol was adapted from *Reniere et al., 2016*. Briefly, 13 µL of overnight culture of *L. monocytogenes* grown in TSB were mixed with 4 mL of molten (55°C) top-agar (0.8% NaCl and 0.8% agar) spread evenly on tryptic soy agar plates. After the agar cooled, Whatman paper disks soaked in 5 µL 1M diamide solution were placed on top of the bacteria-agar. The zone of inhibition (including the disks, diameter 7.5 mm) was measured with a ruler after 18–20 hr of incubation at 37°C.

## *L. monocytogenes* protein aggregation assay

The *L. monocytogenes* aggregation protocol was adapted from Tomoyasu et al with minor modifications (*Tomoyasu et al., 2001*). Colonies of *L. monocytogenes* were picked off a plate and then grown to mid-log in TSB (0.4–0.8 $OD_{600}$). Bacteria were normalized to 0.5 $OD_{600}$ and resuspended in 3 mL of sterile PBS per sample for untreated control, 4-HNE and heat shock. 640 µM 4-HNE was added to the 4-HNE sample and the bacteria were placed at 37°C for an hour. Control heat-shocked bacteria were placed at 37°C for 50 min and then transferred to 56°C for 10 min before being processed identically to 4-HNE and untreated samples. The bacteria were then removed, cooled on ice for 5 min and spun down at 5000 x g for 10 min at 4°C. Pellets were then resuspended in 40 µL buffer A1 (10 mM potassium phosphate buffer, pH 6.5, 1 mM EDTA, 20% w/vol sucrose, 50 units per sample mutanolysin [Sigma, #M9901]) and incubated on ice for 30 min. 360 µL Buffer B1 (10 mM potassium phosphate buffer, pH 6.5, 1 mM EDTA) was added and the sample was sonicated with a microtip sonicator at 40% power, 6 s total, 1 s on, 1 s off. Intact cells were spun out at 2000 x g for 10 min at 4°C and the supernatant was transferred to a fresh tube. Fifty µL of sample supernatant

was taken to measure total protein concentration by BCA assay. The insoluble fraction was isolated by centrifuging the supernatant a 15,000 x g for 20 min at 4°C. The pellets were then frozen and stored at −80C. The pellets were resuspended in 400 μL buffer B1 by brief sonication (40% power, 2 s total, 1 s on, 1 s off) and centrifuged at 15,000 x g for 20 min at 4°C. The washed pellets were then resuspended in 320 μL buffer B1 by brief sonication, after which, 80 μL of 10% v/v IGEPAL was added to remove membrane proteins. The samples were mixed and then centrifuged at 15,000 x g for 30 min at 4°C. This washing procedure was performed twice in total. Finally, the pellets were washed by 400 μL buffer B1 and resuspended in 50 μL PBS by brief sonication. The insoluble protein concentration was measured by microBCA (Thermo, #23235) and the percent of total protein was calculated and plotted.

### *E. coli* protein expression and purification

*rha1*, *rha2*, and *p1-zcr* ORFs were cloned into pET20B expression vectors and transformed into BL21 *E. coli* and grown overnight in LB-ampicillin. The *akr1c1* ORF was cloned into the pET28B expression vector and transformed into Rosette *E. coli* and grown overnight in LB-ampicillin. The overnights were sub-cultured 1:100 in 2L baffled flasks until mid-log (0.4–0.8 $OD_{600}$) after which 0.5 mM IPTG was added. The *pET20b::rha2 E. coli* were induced for 4 hr at 37°C with shaking. to *E. coli pET20B:: rha1, pENT20B::p1-zcr* and *pENT28B::akr1c1* were induced at 17°C for 18 hr with shaking. Immediately prior to induction 0.01% w/v riboflavin was added to *E. coli pET20B::rha1.* Upon induction completion, the bacteria were spun down, resuspended in buffer A (30 mM $K_2HPO_4$, 300 mM NaCl, pH 8) and sonicated on ice with a large sonicator tip at 80% power 1 s on 1 s off for 60 s total. They were then spun down at 15,000 x g for 45 min at 4°C and the supernatant was passed over a nickel resin column (Thermo Fisher Scientific, # PI88222) and eluted using buffer B (30 mM $K_2HPO_4$, 300 mM NaCl, 500 mM Imidazole, pH 8). The final protein was then transferred into PBS using a desalting column (Bio-Rad #7322010). For purification of Rha1, 10 μM FMN (Sigma #F2253) was added at every step of purification.

## Purified enzyme kinetics assessments

Enzyme turnover assays were performed at 37°C in 96-well clear bottom plates (Genesee Scientific, #25–104) in a Synergy HTX plate reader in 200 μL PBS with 20% w/v PEG-8000 using 200 μM NADPH, 0.2 μM enzyme and a range of 4-HNE concentrations from 0 to 0.8 mM. For the aldehyde panel enzyme turnover assessment, 0.64 mM of each aldehyde was used. NADPH consumption was measured at the 340 nm wavelength. Rha1 turnover was performed in the presence of 10 μM FMN.

## Chemical reduction of 4-HNE

One hundred μL of 64 mM 4-HNE was reduced by adding a molar excess (1 mg) of sodium borohydride ($NaBH_4$), which was left to sit at room temperature for an hour. The reaction was quenched for 1 hr at room temperature with 100 μL 1.5% v/v glacial acetic acid in water. The final product of the reaction was 1,4-dihydroxynonene (1,4-DHN) as confirmed by TLC (described below).

## Thin layer chromatography (TLC)

Enzyme turnover assays were performed as described above, except the reactions had final concentrations of 4 μM enzyme, 1.6 mM 4-HNE, and 1.6 mM NADPH. The reactions proceeded for 1 hr at room temperature. Once the reaction was complete, 6 μL of reaction volume was spotted on the bottom of the TLC plate (Millipore Sigma, #105554) and run using a 2:1 mix of diethyl ether:hexanes (Sigma, #296082; Thermo Fisher, #H303). The TLC plate was visualized by dipping the plate in a 10% w/v phosphomolybdic acid (Sigma, #221856) absolute ethanol solution and then vigorously heating the TLC plate on a ceramic hot plate until the appearance of black bands on a light yellow background (15 s to 1 min).

### *B. subtilis* growth curves

*B. subtilis* expressing genes of interest on the pHT01 plasmid (*Nguyen et al., 2007*) were struck on LB-chloramphenicol plates on day one and grown overnight at 30°C. On day 2, colonies were re-struck on LB-chloramphenicol with 1 mM IPTG agar plates overnight at 30°C. On day 3, biomass was scraped and processed as described in the 'bacterial culturing' section above. The bacteria were

then normalized to an $OD_{600}$ of 1 and inoculated 1:100 into a 96-well plate containing TSB chloramphenicol and 0.5 mM IPTG. 4-HNE was then added to the bacteria at various concentrations and the bacteria were allowed to grow at 37°C in Synergy HTX plate reader for 12 hr with shaking.

### 4-HNE survival assays *Bacillus subtilis rha1/2* expression strains

*B. subtilis* were grown and processed as for growth curves described above and then normalized to an $OD_{600}$ of 1 in LB media. The bacteria were then washed twice in sterile PBS and resuspended in sterile PBS. Then the bacteria were diluted 1:100 into sterile PBS and a 160 µM concentration of 4-HNE or mock vehicle (ethanol) was added to the samples. The bacteria were placed at 37°C for an hour, then plated on LB plates. Colonies were enumerated after overnight growth at 30°C.

### *B. subtilis* and *L. monocytogenes* rich media 4-HNE dot blots

*B. subtilis* were grown and processed in the same manner as for the growth curves above. *L. monocytogenes* were processed as described for competition experiments above. Upon $OD_{600}$ normalization to 1, *B. subtilis* were resuspended in TSB-chloramphenicol with 0.5 mM IPTG and 250 µL of this mix was transferred to a sterile Eppendorf tube into which 640 µM 4-HNE was added. *L. monocytogenes* were resuspended in TSB with no antibiotic and no IPTG. The tubes were then incubated at 37°C for 3 hr. At 3 hr, the bacteria were spun down at 17,000 x g for 1 min. The supernatant was then aspirated and the pellet flash frozen in liquid nitrogen. At this point, the bacteria were stored at −80°C until further processing. Once removed from the −80°C, the bacteria were thawed at room temperature and resuspended in 250 µL PBS. The bacteria were then sonicated using a microtip sonicator at 20% power, 1 s on 1 s off for 10 s and placed on ice. The bacteria were then spun at 4°C at 5000 g for 10 min. The subsequent lysate was transferred to fresh Eppendorf tubes containing Halt Proteinase and Phosphatase Inhibitor (Thermo Fisher Scientific, #78442) and stored at −80°C until use. Dot blots were processed as described above.

### *L. monocytogenes* macrophage infection

$0.5 \times 10^6$ primary murine macrophages from WT C57BL/6 mice were plated in BMM media (DMEM with 10% heat inactivated fetal bovine serum, 1 mM L-glutamine, 2 mM sodium pyruvate and 10% L929-conditioned medium) in a tissue culture treated 24-well dish (Greiner Bio, #662165) with the addition of 100 ng recombinant murine IFN-γ (Peprotech, #315–05) for 18 hr. Inoculants of *L. monocytogenes* were statically grown at 30°C overnight, washed twice with sterile PBS and resuspended in PBS before an MOI of 0.1 was added to each macrophage well. The cells were left to sit for an hour after which all wells were washed twice with PBS and gentamicin was added to all but one well, which was lysed in 500 µL water and plated for CFU on LB plates. The remainder of the wells were washed twice with PBS and then lysed and plated for CFU at hours 2, 6, and 9 post-infection.

### *B. subtilis* macrophage survival assay

*B. subtilis* were grown on LB-chloramphenicol with 1 mM IPTG plates and processed into LB chloramphenicol as described above. Once the bacteria were normalized to $OD_{600} = 1$ in LB chloramphenicol, an MOI of 100 was added to $0.5 \times 10^6$ primary murine macrophages from WT C57BL/6 or C57BL/6 deficient for phox (*gp91^{phox-/-}*) mice (The Jackson Laboratory, stock # 002365) that have been activated using 100 ng/well recombinant murine IFN-γ (Peprotech, #315–05) for 18 hr. The cells were spinfected at 200 x g for 5 min. At 1.5 hr, cells were washed 2x with sterile PBS and gentamicin was added to the cells. pBMMs were lysed in 500 µL cold water at hours 2, 6, 7, and 8 and plated for CFU. Colonies were enumerated after overnight growth at 30°C on LB plates.

## Acknowledgements

We thank Aruna Menon, Samantha Hopp and Qing Tang for assistance with experiments, Brian Johnson for performing the histology experiments at the Histology and Imaging Core at the University of Washington, Steven Libby for providing gp91 knockout mice and Peter Lauer and Kevin Lang for providing plasmids. We thank members of the Woodward-Reniere supergroup for helpful discussions, and the Mougous, Reniere and Lagunoff labs for reagents. This material is based upon work supported by PHS NRSA T32GM007270 from NIGMS (to HT and AJP), National Science Foundation

Graduate Research Fellowship Program under grant no. DGE-1256082 (to APM), NIH grant 5T32AI055396 (to RCG), Ruth L Kirschstein Predoctoral Fellowship 1F30CA239659-01A1 (to SZ) and National Institutes of Allergy and Infectious Disease grants R01AI116669 and R21AI127833 (to JJW).

## Additional information

### Funding

| Funder | Grant reference number | Author |
|---|---|---|
| National Institute of General Medical Sciences | PHS NRSA T32GM007270 | Hannah Tabakh Alex J Pollock |
| National Science Foundation | DGE-1256082 | Adelle P McFarland |
| National Institute of Allergy and Infectious Diseases | R01AI116669 | Joshua J Woodward |
| National Institute of Allergy and Infectious Diseases | R21AI127833 | Joshua J Woodward |
| National Cancer Institute | 1F30CA239659-01A1 | Shivam A Zaver |
| National Institute of Allergy and Infectious Diseases | 5T32AI055396 | Rochelle C Glover |

The funders had no role in study design, data collection and interpretation, or the decision to submit the work for publication.

### Author contributions

Hannah Tabakh, Conceptualization, Formal analysis, Funding acquisition, Investigation, Methodology, Writing - original draft; Adelle P McFarland, Conceptualization, Formal analysis, Funding acquisition, Investigation, Methodology, Writing - review and editing; Maureen K Thomason, Investigation, Writing - review and editing; Alex J Pollock, Funding acquisition, Investigation, Methodology, Writing - review and editing; Rochelle C Glover, Shivam A Zaver, Funding acquisition, Investigation, Writing - review and editing; Joshua J Woodward, Conceptualization, Resources, Supervision, Funding acquisition, Writing - original draft, Project administration

### Author ORCIDs

Hannah Tabakh (iD) https://orcid.org/0000-0002-5786-9394
Adelle P McFarland (iD) http://orcid.org/0000-0001-7534-1158
Rochelle C Glover (iD) http://orcid.org/0000-0002-5699-0184
Joshua J Woodward (iD) https://orcid.org/0000-0002-4630-403X

### Ethics

Animal experimentation: All experiments involving mice were performed in compliance with guidelines set by the American Association for Laboratory Animal Science (AALAS) and were approved by the Institutional Animal Care and Use Committee (IACUC) at the University of Washington under protocol #4289-01.

### Decision letter and Author response

Decision letter https://doi.org/10.7554/eLife.59295.sa1
Author response https://doi.org/10.7554/eLife.59295.sa2

## Additional files

### Supplementary files

- Source data 1. Global gene expression of mid-log *L. monocytogenes* treated with 640 μM 4-HNE or ethanol control for 20 min.
- Supplementary file 1. Bacterial strains used in this study.

- Supplementary file 2. Plasmids used in this study.
- Supplementary file 3. Primers used in this study.
- Supplementary file 4. DNA coding sequences used for protein expression and *trans* complementation in this study.
- Transparent reporting form

## Data availability

All RNA sequencing data have been deposited to the GEO and are accessible using accession number GSE150188.

The following dataset was generated:

| Author(s) | Year | Dataset title | Dataset URL | Database and Identifier |
|---|---|---|---|---|
| Tabakh H, Woodward JJ | 2020 | 4-hydroxy-2-nonenal (4-HNE) induced transcriptional changes in Listeria monocytogenes | https://www.ncbi.nlm.nih.gov/geo/query/acc.cgi?acc=GSE150188 | NCBI Gene Expression Omnibus, GSE150188 |

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
