## [Decision Letter]

**Acceptance summary:**

As more is discovered about host-pathogen wars, this study reveals another host
defense mechanism that can be neutralized by an intracellular pathogen. In this
case, the under-appreciated toxic alpha-beta carbonyl 4-hydroxy-2-nonenal that is
produced by ROS-mediated oxidation of host membrane poly-unsaturated fatty acids was
found to be neutralized by two genes from *Listeria monocytogenes*
when expressed in an avirulent bacteria.

**Decision letter after peer review:**

Thank you for submitting your article "4-hydroxy-2-nonenal antimicrobial
toxicity is neutralized by an intracellular pathogen" for consideration by
*eLife*. Your article has been reviewed by 3 peer reviewers, one
of whom is a member of our Board of Reviewing Editors, and the evaluation has been
overseen by Wendy Garrett as the Senior Editor. The following individual involved in
review of your submission has agreed to reveal their identity: Camille Danne
(Reviewer #3).

The reviewers have discussed the reviews with one another and the Reviewing Editor
has drafted this decision to help you prepare a revised submission.

As the editors have judged that your manuscript is of interest but, as described
below, essential additional experiments are required before it is published, we
would like to draw your attention to changes in our revision policy that we have
made in response to COVID-19 (https://elifesciences.org/articles/57162). First,
because many researchers have temporarily lost access to the labs, we will give
authors as much time as they need to submit revised manuscripts. We are also
offering, if you choose, to post the manuscript to bioRxiv (if it is not already
there) along with this decision letter and a formal designation that the manuscript
is "in revision at *eLife*". Please let us know if you
would like to pursue this option or use these reviews to revise and submit
elsewhere.

Summary:

In this work, Tabakh et al. studied the ability of *L. monocytogenes*
to detoxify 4- hydroxy-2-nonenal (4-HNE), a well-studied compound in inflammatory
disease but a rather unexplored metabolite during infection. In this study, 4-HNE is
shown to accumulate in mouse hepatocytes and spleens during infection by *L.
monocytogenes* and is inhibitory to the growth of Firmicutes bacteria.
However, *L. monocytogenes* is somewhat resistant to this compound as
it encodes a detoxification system comprised of rha1 and rha2, which neutralize the
ROS activity of 4- HNE via NADPH dependent reduction.

This study is intriguing, mostly well controlled and potentially opens up a new field
of 4-HNE metabolism in infection. However, the mechanism by which rha1 and 2 mediate
the turnover of 4-HNE needs to be further strengthened wit rigorous biochemistry. In
addition, while the Rha proteins appear to affect 4-HNE, some of the claims need to
be tempered based on the data presented. The most important being that this study
does not convincingly show that 4-HNE production by the host contributes to
*L. monocytogenes* control given that there was no phenotype in
immunocompetent mice for the rha double mutant v. WT *L.
monocytogenes* strains.

Essential revisions:

The following experiments are required for acceptance:

1. Rha1/Rha2 role as NADPH dependent oxidoreductases needs to be further established
with rigorous biochemical studies. The in vitro 4-HNE *L.
monocytogenes* resistance phenotype of the rha1 and rha2 mutants is
somewhat convincing, however the authors should have followed up these results
either by measuring 4-HNE levels or adducts in *L. monocytogenes* or
by better delineating an enzyme mechanism in vitro. These data would be more
supportive of the model that Rha1/2 breaks down 4-HNE or its derivatives in a
physiological setting. It appears Rha1 and especially Rha2 are not very active under
the conditions tested, barely showing a curve representative of catalytic activity.
Characterization of this enzyme requires changes in assay conditions. and more
rigorous analysis.

2. The authors proposed that Rha1/2 could specifically reduce 4-HNE based on
similarity to various enone reductases, but 4-HNE does not contain an enone group.
Also, although it was shown that other aldehydes did not induce rha1/2 expression,
another aldehyde as potential Rha1/2 substrates in this assay could have been tested
to support the specificity claim. Also, is it feasible to measure depletion of 4-HNE
alongside NADPH oxidation over the course of the assay?

3. Are the rha1/rha2 genes conserved? What is the evolutionary picture for these
genes. Some bioinformatics is warranted to understand how prevalent is this
protection mechanism. Are there homologs found in *B. subtilis*? Did
you test 4-HNE toxicity on bacterial species from a different phylum, such as
Bacteroidetes or Proteobacteria? Another intracellular bacteria?

4. Many genes encoding general protein quality control factors were also strongly
induced upon 4-HNE exposure (clpC, groS, dnaK among others). Based on the data
presented, it is possible that Rha1 and Rha2 contribute to *L.
monocytogenes* stress responses by another mechanism and not necessarily
through the specific detoxification of 4-HNE. Does 4-HNE treatment make bacteria
more susceptible to well-characterized proteotoxic stresses like heat shock given
that several well known HSP genes were induced? Does 4-HNE lead to accumulation of
insoluble proteins? How could you explain the interplay between rha1 and rha2? Do
you think they have the same redundant function or rather a complementary function?
If redundant functions, how do you explain that the expression of rha2 in *B.
subtilis* does not provide at least partial resistance to 4-HNE?

5. In Figure 1A, a positive control with direct exposure to 4-HNE should be included
along with infection to provide comparable levels of adduct formation during other
inflammatory diseases.

6. In Figure 4A, a positive control should be included.

7. Figure 2 Show a gene that is not impacted by 4-HNE to highlight the specificity of
the response.

[Editors' note: further revisions were suggested prior to acceptance, as described
below.]

Thank you for resubmitting your work entitled "4-hydroxy-2-nonenal antimicrobial
toxicity is neutralized by an intracellular pathogen" for further consideration
by *eLife*. Your revised article has been evaluated by Wendy Garrett
as the Senior Editor and a Reviewing Editor.

The manuscript has been improved but there are some remaining issues that need to be
addressed, as outlined below:

Overall, the reviewers think the authors addressed the majority of concerns.

However, the text appears to be overstating results and unless complementation
experiments are provided the text must correctly reflect the data. Please revise
text and/or data and highlight modified text/data changes.

For Figure 1, the authors are incorrectly interpreting the histopathology; they claim
the staining of 4HNE is around bacteria but what I am seeing is tons of 4HNE
staining everywhere; The reddish color in the infected tissues clearly shows diffuse
staining everywhere. It doesn't take away from their conclusion at all, they just
need to reword their text to make them not try to claim the staining is only around
the bacteria.

Concerning characterization of bacterial sensitivity to 4-HNE toxicity, authors use
log to describe the reduction in bacterial CFU, except for Lmo where they use %.
Please use consistent labeling of axis so comparisons can be made.

The abstract should be rewritten as the authors say that Rha1 and Rha2 mutations do
not impact Lmo infectious potential, but in the Results section the message is not
that clear. The finding are fine but the conclusions are over stated as it might be
due to redundancy in stress response proteins. The authors need to rephrase to avoid
excessive overstating conclusions.

Finally, the data is not convincing to support the statement that rha genes provide
any fitness advantage in vivo to Lmo as all the other animal data are negative. The
authors would need a complementation assay for panel F to rule out something else
minor like polarity. In the absence of this data, the authors must temper their
conclusions based on that figure. The Bacillus result is more robust if they want to
claim Rha proteins CAN be protective but there is no convincing data to show they
are necessary.

---

## [Author Response]

Essential revisions:The following experiments are required for acceptance:1. Rha1/Rha2 role as NADPH dependent oxidoreductases needs to be further
established with rigorous biochemical studies. The in vitro 4-HNE L.
monocytogenes resistance phenotype of the rha1 and rha2 mutants is somewhat
convincing, however the authors should have followed up these results either by
measuring 4-HNE levels or adducts in L. monocytogenes or by better delineating
an enzyme mechanism in vitro. These data would be more supportive of the model
that Rha1/2 breaks down 4-HNE or its derivatives in a physiological setting. It
appears Rha1 and especially Rha2 are not very active under the conditions
tested, barely showing a curve representative of catalytic activity.
Characterization of this enzyme requires changes in assay conditions. and more
rigorous analysis.“…however the authors should have followed up these results either by measuring
4-HNE levels or adducts in L. monocytogenes…”

We thank the reviewers for their suggestion. We have now added new data in Figure 4D
to the manuscript showing a ~50% increase in 4-HNE adducts in *L.
monocytogenes∆∆rha1/2* mutant compared to WT upon incubation with 640µM
4-HNE by dot blot and that complementation with either gene had modest effects.
Given the nominal magnitude of adduct accumulation it is difficult to draw much from
this data. Because expression of these genes in *B. subtilis* had
much larger impacts on adduct accumulation, these findings suggest that other
mechanisms for clearing 4-HNE adducted proteins may be present in
*Listeria*, which may mask the effects of Rha1/2. This is also
consistent with the significant level of resistance that *Listeria*
has over *B. subtilis* even when the latter organism expresses
Rha1/2. Clearly Rha1/2 are only the beginning of the story for 4-HNE resistance in
*Listeria* and future studies are warranted to explore this
resistance in further detail.

“…by better delineating an enzyme mechanism in vitro.”

We agreed with the reviewers about our initial enzymatic data. As such, we assessed
purified enzyme turnover in numerous conditions, including in bacterial lysates, in
sterile bacterial media, with the addition of various salts, metals, crowding
agents, and different buffers. From this comprehensive screening of in vitro
biochemical conditions, we found that the addition of 20% PEG 8000 to the reactions
significantly decreased the Km of both Rha1 and Rha2 to within the broad
physiological range of 4-HNE (Rha1 134 µM, Rha2 367 µM). We have added these data as
new Figure 5B. We suspect PEG-induced crowding is decreasing the Km of both the
enzymes, either by shifting the protein to its predicted active dimer form (Rha1) or
perhaps by generally reducing protein instability and therefore increasing the
potential for successful 4-HNE-protein association (Ma and Nussinov, 2013). To
provide a comparison to a known 4-HNE metabolizing enzyme, we tested turnover of the
known human 4-HNE NADPH-dependent detoxification enzyme AKR1C1 under the same
conditions. While AKR1C1 exhibits a lower Km, the turnover rates of Rha1/2 (kcat)
are higher than the mammalian counterpart under these conditions. These data are now
presented as new Figure 5C.

2. The authors proposed that Rha1/2 could specifically reduce 4-HNE based on
similarity to various enone reductases, but 4-HNE does not contain an enone
group.

Although the reviewers are correct in noting that 4-HNE does not contain an enone
group, it does contain an enal group, which is often reduced by a similar enzymatic
mechanism to enone reduction. Ene-reductases specifically have reduction activity
against carbon-carbon double bonds both in enal and enone contexts and we have tried
to clarify this in the text (Toogood and Scrutton, 2018).

Also, although it was shown that other aldehydes did not induce rha1/2
expression, another aldehyde as potential Rha1/2 substrates in this assay could
have been tested to support the specificity claim.

As suggested by the reviewers, we expanded the panel of aldehydes we tested as
potential substrates for both Rha1 and Rha2. Our findings largely mirrored
observations of *rha1* and *rha2* gene induction by
this same panel (Figure 3B), in which activity was not observed for any combination
with one exception, Rha2 exhibited some turnover capacity with the lipid oxidation
product of n-3 fatty acids 4-hydroxy 2hexenal. These observations are in line with
our assignment of Rha1/2 as reductases of the αβ-double bond of 4-HNE, which is
absent in other aldehydes tested in this panel. Collectively these observations are
consistent with these genes functioning in response to enal induced stress and
toxicity and that perhaps Rha2 is more promiscuous than Rha1. Of course, the
conclusions of specificity for 4-HNE are limited by the substrates tested and we
have toned down the language relating to specificity. In fact, we hypothesize that
these enzymes may indeed have other αβ-carbonyl containing substrates, either
endogenous (i.e. quinones) or other αβ-unsaturated compounds that induce
electrophilic stress. These data are now presented as new Figure 5D.

Also, is it feasible to measure depletion of 4-HNE alongside NADPH oxidation over
the course of the assay?

We successfully visualized Rha1 and Rha2 enzymatic conversion of 4-HNE using
thin-layer chromatography with phosphomolybdic acid visualization. These data are
now presented in new Figure 5E. As a control for direct comparison on the TLC plate,
we included AKR1C1, known to generate the alcohol 1,4-dihydroxynonene (1,4-DHN) and
P1-ZCr, an *Arabidopsis thaliana* enzyme known to perform a
hydrogenation reaction to saturate 4-HNE to 4-hydroxynananal (4-HNA). We found 4-HNE
was converted to 1,4-DHN by AKR1C1 as reported, to 4-HNA by P1-ZCr as reported and
both Rha1 and Rha2 enzymes converted 4-HNE to 4-HNA. These studies were not done in
parallel with NADPH consumption due to technical limitations. Specifically, the
amount of enzyme and NADPH in the experiments for monitoring NADPH consumption is
not sufficient to detect the 4-HNE and reaction product 4-HNA using TLC. As such,
the two experiments were performed with different amounts of enzyme and NADPH.
However, we found no 4-HNE conversion with any enzymes in the absence of NADPH.
Overall, we were able to successfully demonstrate NADPH-dependent 4-HNE reduction to
4HNA by both Rha1 and Rha2.

3. Are the rha1/rha2 genes conserved? What is the evolutionary picture for these
genes. Some bioinformatics is warranted to understand how prevalent is this
protection mechanism.

We performed sequence homology analysis for both *rha1* and
*rha2* across the set of bacteria we tested for 4-HNE sensitivity
to obtain a sense of the distribution of both genes across a variety of prokaryotes.
Author response table 1 reports
the sequence identity of the closest matching Rha1 and Rha2-like proteins from
tested organisms. *E. faecalis*, which we found to be both the second
most 4-HNE tolerant bacteria we tested, had the closest homologs to both Rha1 and
Rha2. Among other organisms, we found that the association between 4-HNE survival
and closeness or presence of protein homologs was weak. This is not surprising, as
functional homology and sequence homology are often not very strongly correlated
among oxidoreductase enzymes (Todd, Orengo and Thornton, 2001). This is due to the
fact that very small changes in the binding pocket completely alters substrate
specificity and it’s very difficult, if not impossible, to predict with any
confidence the substrate of many oxidoreductase enzymes from simple sequence
homology. We speculate that some close relatives of *L.
monocytogenes*, including perhaps *E. faecalis*, may
utilize similar mechanisms of 4-HNE detoxification and that other bacteria that are
distantly related, like *P. aeruginosa* likely have other enzymes
with limited sequence homology to Rha1/2 that are capable of performing similar
functions. Rigorous transcriptional, biochemical and other experimental analysis
will be required to say with any confidence if other bacteria code for enzymes with
similar 4-HNE detoxification roles.

**Author response table 1. resptable1:** 

Organism	Rha1 identity (%)	Rha2 identity (%)
Pa	29	31
SA	Absent	32
Ef	59	71
Ec	Absent	34
Fn	24	33
Bs	58	29

Are there homologs found in *B. subtilis*? Did you test 4-HNE
toxicity on bacterial species from a different phylum, such as Bacteroidetes or
Proteobacteria? Another intracellular bacteria?

We thank the reviewer for their suggestions. To address this point, we expanded our
killing assay to include a set of Gram-negative bacteria: *E. coli*
DH10b (K12)*, Pseudomonas aeruginosa* PA01 and *Francisella
novicida* U112, which is also an intracellular pathogen. These data are
presented as new Figure 2A. We found that even among this expanded bacterial cohort,
*L. monocytogenes* still showed the highest survival
capabilities. Interestingly, *F. novicida* was uniquely sensitive to
4-HNE toxicity. Reports that *F. novicida* actively blocks ROS
generating pathways during infection (Mohapatra et al., 2010) suggests that this
organism may avoid 4-HNE mediated toxicity by halting its production rather than
specific detoxification or resistance mechanisms. We have added discussion of these
points to the Results and Discussion sections.

4. Many genes encoding general protein quality control factors were also strongly
induced upon 4-HNE exposure (clpC, groS, dnaK among others). Based on the data
presented, it is possible that Rha1 and Rha2 contribute to L. monocytogenes
stress responses by another mechanism and not necessarily through the specific
detoxification of 4-HNE.

Induction of heat shock proteins in response to electrophilic stress has been
previously reported in *B. subtilis* (Huyen et al., 2009). We agree
with the reviewers that while it is still formally possible that Rha1/2 contribute
to the *Listeria* stress response through an indirect pathway to
4-HNE resistance rather than direct detoxification, our improved in vitro enzymology
(new Figure 5) suggests the most parsimonious explanation is that they play a role
in directly de-toxifying 4-HNE. However, to provide a more comprehensive picture of
Rha1 and Rha2 expression in bacterial stress responses, we measured
*rha1/2* expression in response to a panel of non-aldehyde
stressors, including heat, diamide induced disulfide stress, and the RNS agent
nitric oxide. We found that NO was unable to induce either gene, but that heat did
induce both genes by approximately 30-fold, and the disulfide stress agent diamide
induced *rha1* by 11-fold and *rha2* by 100-fold.
However, the most robust induction of both *rha1* and
*rha2* was with 4-HNE, as we observed previously with our
aldehyde panel in Figure 3B. Induction by heat and diamide suggests that
*rha1/2* are induced during other stress responses in *L.
monocytogenes*. Despite their induction under both heat and diamide
induced thiol stress, loss of each gene had no discernible impact on bacterial
survival in these conditions. These data are now presented in Figure 3C and Figure
4B,C.

Does 4-HNE treatment make bacteria more susceptible to well-characterized
proteotoxic stresses like heat shock given that several well known HSP genes
were induced?

We found that indeed, 4-HNE treatment prior to heat shock (50°C for 10 minutes) does
increase bacterial death. This is consistent with 4-HNE impacting proteotoxic stress
responses. This data is now presented in Figure 2E. Given that many of the proteases
involved in the heat shock require active site nucleophiles that are likely highly
susceptible to 4-HNE adduction, it is feasible that 4-HNE poisons these proteins
from functioning to clear damaged proteins during elevated temperature. While these
observations are intriguing, detailing the mechanism of this synergy is beyond the
scope of this manuscript.

Does 4-HNE lead to accumulation of insoluble proteins?

We found that in WT *L. monocytogenes* 4-HNE does not lead to
significant total protein aggregation compared to untreated control, especially
compared to the positive control of elevated temperature exposure. These data are
now presented in Figure 4E. In addition, the *∆∆rha1/2* mutant does
appear to have only a modest (10-15%) and not statistically significant increase in
protein aggregation accumulation with 4-HNE treatment compared to WT. This suggests
that the difference in WT versus *∆∆rha1/2* survival in 4-HNE is not
driven by global adduct accumulation or protein aggregation but rather other
molecular targets susceptible to 4-HNE reactivity. Such targets may be other
macromolecules (i.e. lipids, nucleic acids, etc) or adduction to specific cellular
proteins, namely inactivation of specific essential proteins that are not reflected
in the global analyses conducted here.

How could you explain the interplay between rha1 and rha2? Do you think they have
the same redundant function or rather a complementary function? If redundant
functions, how do you explain that the expression of rha2 in *B.
subtilis* does not provide at least partial resistance to 4-HNE?

Although we do not know exactly what the relationship between the two proteins is, we
suspect that the two enzymes serve complimentary functions. Both enzymes reduce
4-HNE to 4-HNA (new Figure 5E) however, they appear to function non-redundantly,
with both enzymes being expressed heterologously in *B. subtilis*
allowing for greater than simply additive survival for the expressing *B.
subtilis* (Supplemental Figure 3B, Figure 6A). The enzymes have
different Km affinities for 4-HNE (Figure 5)

as well as different 4-HNE turnover specificity. Rha1 is seemingly specific for 4-HNE
as a substrate for NADPH oxidation while Rha2 is able to perform the reduction of
both 4-HNE and 4-HHE (Figure 5D). The differential induction of
*rha1* versus *rha2* with our aldehyde and stress
panel (Figure 3B-C; see our response to point 4 above) suggests that the two
proteins may be regulated by different inputs or play specific roles during 4-HNE
exposure possibly through distinct protein localization within the cell.

5. In Figure 1A, a positive control with direct exposure to 4-HNE should be
included along with infection to provide comparable levels of adduct formation
during other inflammatory diseases.

We repeated the TIB73 *L. monocytogenes* infection experiments with
the included positive control of TIB73s that were treated for 10 minutes with 10µM
4-HNE in PBS. These data are presented in new Figure 1A. We found that the treated
cells accumulate adducts at a similar level to 6 hours of *L.
monocytogenes* infection. Although that suggests overall low levels of
4-HNE, we believe that this is due to the combination of (1) a small amount of cells
infected with *L. monocytogenes* at 6 hours post infection and (2)
the highly segregated nature of 4-HNE accumulation. We believe that 10µM overall
4HNE accumulation masks the much higher localized levels of 4-HNE in cells.
Additionally, measurement of 4-HNE levels in cells actively producing the
metabolite, as during infection, reflects the amount of protein conjugates that are
accumulating and being degraded.

6. In Figure 4A, a positive control should be included.

We included the positive control of the human aldo-keto reductase AKR1C1 that is
well-known to perform NADPH-dependent oxidation of 4-HNE. This data is now in the
new Figure 5C.

7. Figure 2 Show a gene that is not impacted by 4-HNE to highlight the
specificity of the response.

We included the gene *rplD* whose expression we found to not be
impacted by 4-HNE exposure. This data is now in new Figure 2D.

[Editors' note: further revisions were suggested prior to acceptance, as described
below.]

The manuscript has been improved but there are some remaining issues that need to
be addressed, as outlined below:Overall, the reviewers think the authors addressed the majority of concerns.However, the text appears to be overstating results and unless complementation
experiments are provided the text must correctly reflect the data. Please revise
text and/or data and highlight modified text/data changes.For Figure 1, the authors are incorrectly interpreting the histopathology; they
claim the staining of 4HNE is around bacteria but what I am seeing is tons of
4HNE staining everywhere; The reddish color in the infected tissues clearly
shows diffuse staining everywhere. It doesn't take away from their conclusion at
all, they just need to reword their text to make them not try to claim the
staining is only around the bacteria.

We completely agree that there is an increase in 4-HNE staining throughout the spleen
following *L. monocytogenes* infection as indicated by the overall
reddish color between the infected and uninfected tissue sections shown in Figure 1B
and C, respectively. With increased magnification as shown in Figure 1E, there are
clearly regions of the tissue section that stand out as having a much darker
staining than the diffuse background seen in much of the tissue section. These cells
have such a strong signal for the 4HNE that they are very dark brown. We have added
arrows to Figure E to highlight these regions. In comparison to the *L.
monocytogenes* imaging in Figure 1D at the same magnification, the
staining pattern looks similar. At 100X magnification, the 4-HNE labeled cells
exhibiting the darkest staining have a punctate pattern (Figure 1G) that looks
strikingly similar to the *L. monocytogenes* staining (Figure 1F). We
completely agree that this comparison of staining is clearly not sufficient evidence
to conclude that the bacteria are being directly labeled by the metabolite. It is
equally possible that there is a subset of spleen cells that simply produce
significantly elevated levels of 4-HNE. As such, we have removed the statement that
this increased staining is around the bacteria (Lines 106-109) and added arrows to
Figure 1E to further direct the reader’s attention to the enhanced staining in
specific regions of the tissue.

Concerning characterization of bacterial sensitivity to 4-HNE toxicity, authors
use log to describe the reduction in bacterial CFU, except for Lmo where they
use %. Please use consistent labeling of axis so comparisons can be made.

We have changed the wording on Lines 130-131 to use log reduction to describe
*L. monocytogenes* viability following 4-HNE exposure.

The abstract should be rewritten as the authors say that Rha1 and Rha2 mutations
do not impact Lmo infectious potential, but in the Results section the message
is not that clear. The finding are fine but the conclusions are over stated as
it might be due to redundancy in stress response proteins. The authors need to
rephrase to avoid excessive overstating conclusions.

We have included a statement highlighting that Rha1 and Rha2 are not necessary for
*L. monocytogenes* infectious potential (Lines 22-23).

Finally, the data is not convincing to support the statement that rha genes
provide any fitness advantage in vivo to Lmo as all the other animal data are
negative. The authors would need a complementation assay for panel F to rule out
something else minor like polarity. In the absence of this data, the authors
must temper their conclusions based on that figure. The Bacillus result is more
robust if they want to claim Rha proteins CAN be protective but there is no
convincing data to show they are necessary.

We agree that the reduction in CFU following macrophage infection is not sufficient
to make any claim about in vivo fitness associated with these genes in *L.
monocytogenes*. We have made changes to the text to reflect this (Lines
240-244).